# MindSimulator: Exploring Brain Concept Localization via Synthetic FMRI

**Guangyin Bao**[1], **Qi Zhang**[1], **Zixuan Gong**[1], **Zhuojia Wu**[1], **Duoqian Miao**[1]*
[1]Tongji University

## Abstract

Concept-selective regions within the human cerebral cortex exhibit significant activation in response to specific visual stimuli associated with particular concepts. Precisely localizing these regions stands as a crucial long-term goal in neuroscience to grasp essential brain functions and mechanisms. Conventional experiment-driven approaches hinge on manually constructed visual stimulus collections and corresponding brain activity recordings, constraining the support and coverage of concept localization. Additionally, these stimuli often consist of concept objects in unnatural contexts and are potentially biased by subjective preferences, thus prompting concerns about the validity and generalizability of the identified regions. To address these limitations, we propose a data-driven exploration approach. By synthesizing extensive brain activity recordings, we statistically localize various concept-selective regions. Our proposed *MindSimulator* leverages advanced generative technologies to learn the probability distribution of brain activity conditioned on concept-oriented visual stimuli. This enables the creation of simulated brain recordings that reflect real neural response patterns. Using the synthetic recordings, we successfully localize several well-studied concept-selective regions and validate them against empirical findings, achieving promising prediction accuracy. The feasibility opens avenues for exploring novel concept-selective regions and provides prior hypotheses for future neuroscience research.

## 1 Introduction

The human brain's visual cortex is decisive in processing and perceiving visual information. Neuroscience researchers have long dedicated themselves to unraveling the brain's visual mechanisms, making impressive strides such as in brain visual encoding (Mitchell et al., 2008), decoding (Gong et al., 2024b), and visual perception (Chen et al., 2020). However, the process of forming visual cognition remains to be explored. Notably, localizing the various functional organizations and activation patterns of the visual cortex that correspond to human conceptual cognition is considered pivotal yet remains a challenging frontier (Huth et al., 2016; Henderson et al., 2023; Luo et al., 2024). Numerous neuroscience studies have illustrated that specific regions of the visual cortex exhibit concept selectivity. When individuals receive visual stimuli related to particular concepts (such as places, bodies, faces, words, colors, and foods), the respective cortical regions exhibit significant activation (Epstein & Kanwisher, 1998; Sergent et al., 1992; Jain et al., 2023; Pennock et al., 2023; Kanwisher et al., 1997; Allen et al., 2022). These regions are termed visual concept-selective regions and play a vital role in advancing the understanding of brain visual cognition.

Typically, identifying concept-selective regions relies on the functional localizer (fLoc) experiments (Stigliani et al., 2015; Allen et al., 2022). To this end, neuroscience researchers need to purposefully and manually construct visual stimulus sets associated with specific visual concepts. These stimuli are subsequently presented to subjects for costly functional magnetic resonance imaging (fMRI) scans, aiming to localize these regions by statistically analyzing the fMRI data related to visual stimuli. However, this experiment-driven exploration encounters three major limitations: 1) Real fMRI-image data are scarce, resulting in concept-selective region localization being limited

---

*Corresponding author.

to a few concept categories. 2) The collection of visual stimuli accompanied by an artificial selection is biased. 3) Existing manual-constructed visual stimuli sets often consist of isolated objects in unnatural scenes. These limitations naturally prompt concerns about the generalizability of visual concepts of concept localization. To overcome these issues, we intend to leverage a flexible data-driven approach to break through the limitations of manual-constructed stimuli and expensive experimental fMRI collection to locate more generalized and precise concept-selective regions.

In this paper, we propose *MindSimulator*, a novel generative fMRI encoding model for flexibly synthesizing individual fMRI corresponding to concept-oriented visual stimuli. *MindSimulator* operates through a reverse process in conjunction with fMRI decoding (Naselaris et al., 2011). Building on the significant advancements in fMRI visual decoding (Scotti et al., 2024a;b; Gong et al., 2024a; Shen et al., 2024), we are motivated to develop powerful fMRI encoding models by leveraging existing fMRI datasets (Allen et al., 2022) and advanced generative deep learning techniques (Ho et al., 2020; Rombach et al., 2022). Specifically, *MindSimulator* first constructs an fMRI autoencoder and aligns fMRI latent space with well-trained visual stimuli (i.e., image) representation space. Subsequently, a diffusion model is integrated to learn fMRI's conditional probability distribution for a given concept-oriented visual stimuli on the fMRI-image joint representation space. Once *MindSimulator* is trained effectively, it serves as an individual's brain capable of generating fMRI data corresponding to diverse concept-related visual stimuli ideally.

In addition, we evaluate the fidelity of synthetic fMRI at the voxel level and semantic level, ensuring that *MindSimulator* can maximally restore recognizable neural response patterns, i.e., visual semantic contained in fMRI. More importantly, *MindSimulator* experimentally shows excellent generalization capability, even for out-of-distribution visual stimuli, thereby enabling synthesizing extensive fMRI of various concepts and achieving an expansion for scarce fMRI data.

On this basis, we use synthetic fMRI to localize concept-selective regions. Statistically, the data-driven localization enables us to explore concept-selective regions across various categories, facilitating more finer-grained region discovery, instead of being limited to the categories present in the fLoc experiment's stimuli categories. We conduct concept-selective region localization experiments using fMRI synthesized by *MindSimulator* and verify its feasibility by predicting the existing regions empirically delineated by the fLoc experiments.

## 2 RELATED WORKS

**FMRI Encoding.** The fMRI encoding research has been explored over a long period (Mitchell et al., 2008; Huth et al., 2016; Gu et al., 2022; Tang et al., 2023). Existing approaches used regression models to map image features to voxel space. Some researchers focused on selecting better image features (Han et al., 2019; Wang et al., 2023) or better visual stimuli (Luo et al., 2024), and simple linear regression was used for better interpretability. The others were concerned with developing better regression models (Gifford et al., 2023; Yang et al., 2023; Adeli et al., 2023; Ma et al., 2024; Liang et al., 2024; Beliy et al., 2024). We are the first to develop a generative encoding model.

**Generative Models.** Generative models can sample from noise to generate data with clear semantics. Mainstream generative architectures include variational autoencoders (Kingma, 2013; Van Den Oord et al., 2017), generative adversarial networks (Goodfellow et al., 2014), and diffusion models (Sohl-Dickstein et al., 2015; Ho et al., 2020; Song et al., 2020). Generative models conditional on images can output text (Li et al., 2022; 2023), images (Xu et al., 2023; Zhang et al., 2023), or videos (Blattmann et al., 2023; Shi et al., 2024).

**FMRI Visual Semantic Decoding.** Advanced brain decoding studies have been able to decipher clear visual semantics from brain activity fMRI recordings and restore seen visual scenes (Lin et al., 2022; Chen et al., 2023; Scotti et al., 2024a;b; Wang et al., 2024; Quan et al., 2024; Gong et al., 2024a;b; Shen et al., 2024; Bao et al., 2024; Xia et al., 2024; Chen et al., 2024), demonstrating their strong capability for recognizing global neural response patterns. In our study, these pre-trained visual decoding models are components of our semantic-level evaluation pipeline.

## 3 METHOD

Our *MindSimulator* aims to map arbitrary visual stimuli into individual brain activity using paired natural image stimuli and fMRI recordings. We begin by articulating our motivation for adopting the

generative architecture and provide a general overview of the proposed method. We then describe each key component of *MindSimulator*. Finally, we describe how noise can be translated into cortical activity fMRI recordings conditioned on given visual stimuli.

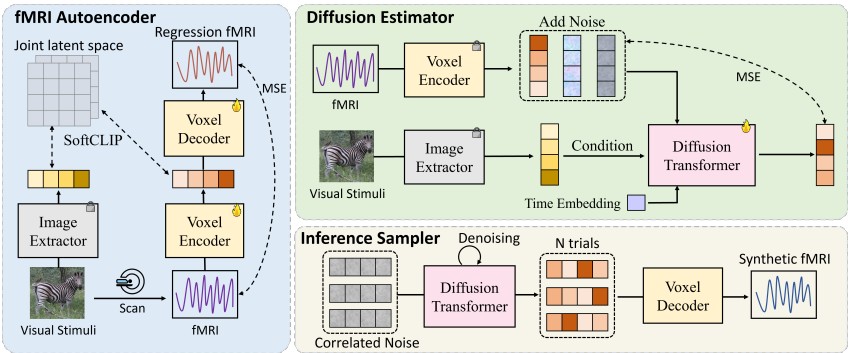

Figure 1: Overview of the proposed MindSimulator. It comprises a fMRI autoencoder, a Diffusion Estimator, and a Inference Sampler. The fMRI autoencoder enables mutual transformation between voxels and fMRI representations. The diffusion estimator generates fMRI from noise conditioned on images. The inference sampler achieves high-precision fMRI synthesis. Please refer to Sections 3.2 to 3.4 for more details.

## 3.1 MOTIVATION AND OVERVIEW

At least two types of approaches can be used for mapping visual stimuli or their representations to corresponding fMRI recordings. The regression models directly map visual stimuli to target fMRI recordings through a deep network; whereas the generative models use visual stimuli as conditional guidance to generate the target fMRI recordings from Gaussian noise. However, observations show that there are noticeable differences in brain activity fMRI recordings even when receiving the same visual stimuli (Horikawa & Kamitani, 2017; Allen et al., 2022), suggesting that visual stimuli and fMRI recordings essentially exhibit a one-to-many correspondence. Regression models fail to capture this phenomenon because the mapped fMRI recording of a given visual stimuli is unique; on the contrary, generative models treat differential fMRI recordings as outcomes of random sampling from a conditional probability distribution based on given visual stimuli. Therefore, the mechanism of generative models aligns more closely with the behavioral performance of the human brain.

The proposed *MindSimulator* adopts a generative architecture. As illustrated in Figure 1, it consists of three components: 1) a **fMRI Autoencoder**, which facilitates the interconversion between fMRI voxel and its high-dimensional representation; 2) a **Diffusion Estimator**, which learns the conditional distribution of fMRI based on given visual stimuli; and 3) a **Inference Sampler**, which generates accurate synthetic fMRI using multi-trial enhancement and correlated noise.

## 3.2 FMRI AUTOENCODER

The low signal-to-noise fMRI involves complex brain activity, bringing difficulty to estimating its data distribution. Thus, we project raw fMRI into high-dimensional latent representation via the fMRI autoencoder and estimate its conditional distribution in the latent space instead. Specifically, sampling a paired training data $(x, y)$ from subject-individual fMRI dataset $\mathcal{S}$, where $x \in \mathbb{R}^l$ denotes preprocessed fMRI blood oxygenation level-dependent (BOLD) voxels and $y$ denotes the corresponding visual stimuli. The autoencoder consists of a voxel encoder $\mathcal{E}(\cdot)$ and a voxel decoder $\mathcal{D}(\cdot)$. The voxel encoder embeds $x$ to a high-dimensional fMRI representation $\mathcal{X} = \mathcal{E}(x) \in \mathbb{R}^{m \times d}$, resulting in $m$ $d$-dimensional tokens. The voxel decoder works just the opposite, decoding the high-dimensional voxel representation $\mathcal{X}$ back to fMRI voxels, i.e. $\hat{x} = \mathcal{D}(\mathcal{X}) \in \mathbb{R}^l$. Finally, we train this autoencoder using a strong voxel-wise supervision objective: $\mathcal{L}_{\text{mse}} = \mathbb{E}_{x \sim \mathcal{S}} ||x - \hat{x}||_2^2$. To simplify, we omit the notation related to the mini-batch sampling.

Inspired by existing text-to-image generative models (Rombach et al., 2022; Saharia et al., 2022), we further construct cross-modal joint latent spaces to facilitate stable convergence of generative models. Therefore, we align the fMRI representation space with a pre-train image representation space. Specifically, we use trained CLIP ViT (Radford et al., 2021) as the image extractor $\mathcal{V}(\cdot)$

for visual stimuli. Note that the previous study has shown that image representations obtained by contrastive learning are more suitable for fMRI encoding task (Wang et al., 2023). The image representation $\mathcal{Y} = \mathcal{V}(y) \in \mathbb{R}^{m \times d}$ has consistent dimension with fMRI representation $\mathcal{X}$. Subsequently, we use SoftCLIP loss (Gao et al., 2024) with a cosine-scheduled temperature factor $\tau$ to supervise the cross-modal alignment process:

$$
\begin{aligned}
\mathcal{L}_{\text{softclip}} = &-\frac{1}{|\mathcal{S}|} \sum_{i=1}^{|\mathcal{S}|} \sum_{j=1}^{|\mathcal{S}|} \left[ \frac{\exp(\mathcal{X}_i \cdot \mathcal{X}_j / \tau)}{\sum_{k=1}^{|\mathcal{S}|} \exp(\mathcal{X}_i \cdot \mathcal{X}_k / \tau)} \cdot \log \left( \frac{\exp(\mathcal{Y}_i \cdot \mathcal{X}_j / \tau)}{\sum_{k=1}^{|\mathcal{S}|} \exp(\mathcal{Y}_i \cdot \mathcal{X}_k / \tau)} \right) \right] \\
&-\frac{1}{|\mathcal{S}|} \sum_{i=1}^{|\mathcal{S}|} \sum_{j=1}^{|\mathcal{S}|} \left[ \frac{\exp(\mathcal{Y}_i \cdot \mathcal{Y}_j / \tau)}{\sum_{k=1}^{|\mathcal{S}|} \exp(\mathcal{Y}_i \cdot \mathcal{Y}_k / \tau)} \cdot \log \left( \frac{\exp(\mathcal{X}_i \cdot \mathcal{Y}_j / \tau)}{\sum_{k=1}^{|\mathcal{S}|} \exp(\mathcal{X}_i \cdot \mathcal{Y}_k / \tau)} \right) \right] .
\end{aligned}
\tag{1}
$$

We fix the pre-trained visual extractor and train the fMRI autoencoder end-to-end with a joint loss:

$$
\mathcal{L}_{\text{autoencoder}} = \mathcal{L}_{\text{mse}} + \mathcal{L}_{\text{softclip}} .
\tag{2}
$$

### 3.3 Diffusion Estimator

Contrastive learning facilitates disjointed cross-modal representations (Ramesh et al., 2022). Accordingly, the fMRI representation $\mathcal{X}$ output by the voxel encoder is parallel with the corresponding image representation $\mathcal{Y}$ while keeping a certain distance. On this basis, we train a generative model to learn the conditional probability distribution of fMRI representation on a given image representation. Previous study (Oko et al., 2023) has shown that diffusion models are suitable to achieve the goal. Accordingly, we construct the diffusion estimator $\mathcal{P}(\cdot)$ with $T$ timesteps. In the estimator, by applying the reparameterization trick, the noised fMRI representation $\mathcal{Z}_t^{\mathcal{X}}$ can be formalized as:

$$
\mathcal{Z}_t^{\mathcal{X}} = \sqrt{\bar{\alpha_t}} \cdot \mathcal{X} + \sqrt{1 - \bar{\alpha_t}} \cdot \epsilon, \quad \bar{\alpha}_t = \prod_{m=1}^{t} \alpha_m, \quad t \sim [\, 1, 2, \cdots, T \,] ,
\tag{3}
$$

where $\alpha_m$ denotes the noise schedule hyperparameter and $\epsilon \sim \mathcal{N}(0, 1)$ is Gaussian noise. Unlike common diffusion models that predict noise (Ho et al., 2020), we aim to learn the conditional distribution such that our diffusion estimator directly predicts target fMRI representations. Using $\mathcal{T}_t$ to denote learnable time embedding of timestep $t$, its learning objective can be formalized as:

$$
\mathcal{L}_{\text{diffusion}} = \mathbb{E}_{\epsilon, t, (x,y) \sim \mathcal{S}} [\, ||\mathcal{P}(\mathcal{Z}_t^{\mathcal{X}}, \mathcal{Y}, \mathcal{T}_t) - \mathcal{X}||_2^2 \,] .
\tag{4}
$$

Here, we adopt a Transformer architecture (Vaswani et al., 2017; Peebles & Xie, 2023) for diffusion estimator $\mathcal{P}(\cdot)$, integrating image representations as conditions through cross-attention modules.

### 3.4 Inference Sampler

Once the diffusion estimator is trained, it can progressively predict target fMRI representation $\hat{\mathcal{X}}$ conditional on the given image representation, formalized as:

$$
\hat{\mathcal{Z}}_{t-1}^{\mathcal{X}} = \mathcal{P}(\hat{\mathcal{Z}}_t^{\mathcal{X}}, \mathcal{Y}, \mathcal{T}_t), \quad \hat{\mathcal{Z}}_T^{\mathcal{X}} \sim \mathcal{N}(0, 1), \quad \hat{\mathcal{X}} = \hat{\mathcal{Z}}_0^{\mathcal{X}} .
\tag{5}
$$

To acquire more accurate synthetic brain activity fMRI, we introduce the following two strategies to improve synthesis performance.

**Multi-Trial Enhancement.** In neuroscience, intra-subject voxel-wise reproducibility is crucial for experimental exploration. Specifically, it involves correlating fMRI data across multiple trials of the same visual stimuli to ensure the precision of neural responses. Inspired by this treatment, we consider the reproducibility of multiple generated fMRI. We generate $N$ fMRI from $N$ different Gaussian noise, simulating the viewing of an image $N$ times. These synthetic fMRI correspond to the same visual stimuli. Finally, we average synthetic fMRI to achieve a more accurate generation.

**Correlated Gaussian Noise.** We target synthetic fMRI to exhibit a high correlation in neural response patterns across the $N$ trials, as lower variance typically leads to increased fMRI synthesis performance. It is well known that the uncertainty in the generated results arises from the randomness of the Gaussian noise; thus, we propose to use correlated Gaussian noise as the input for $N$-trial

generation. To create $N$ correlated Gaussian noise, we first randomly sample two independent noise $\epsilon_1 \sim \mathcal{N}(0, 1)$ and $\epsilon_2 \sim \mathcal{N}(0, 1)$. Then, we apply weights and thereby obtain $N$ new noise $\epsilon_n$:

$$\epsilon_n = \sqrt{\beta_n} \cdot \epsilon_1 + \sqrt{1 - \beta_n} \cdot \epsilon_2, \ \ n \in [1, 2, \cdots, N] \tag{6}$$

By setting different weights $\beta_n$, we can obtain a series of correlated standard Gaussian noises, which are located on the curve between $\epsilon_1$ and $\epsilon_2$ in the high-dimensional space and have high similarity.

## 4 EXPERIMENTS SETUP

### 4.1 DATASETS

We use the Natural Scenes Dataset (NSD) (Allen et al., 2022), which is an extensive whole-brain fMRI dataset gathered from 8 subjects viewing images from MSCOCO (Lin et al., 2014). In NSD experiments, participants were required to view 10,000 images for 3 trials, thereby acquiring 30,000 fMRI scans. Our evaluation focuses on Subj01, Subj02, Subj05, and Subj07 because these subjects completed all experiment sessions. The ~9,000 unique images for each subject are used for training and the remaining ~1,000 shared images are used for evaluation. During the training phase, all three fMRI of the same image are used individually; while for testing, three repeats are averaged. We use beta-activations computed using GLMSingle (Prince et al., 2022) and normalize each voxel to $\mu = 0$, $\theta = 1$ on a per-session basis. We average multi-trail voxels for the testing set. Since we are focusing on the visual cortex regions, we apply the official *nsdgeneral* region-of-interest (ROI) mask, which spans visual regions ranging from the early visual cortex to higher visual areas. We flatten the fMRI selected by ROI and then obtain one-dimensional voxel sequences for encoding.

### 4.2 IMPLEMENTATION DETAILS

For the image extractor, we used pre-trained CLIP ViT-L/14. It encodes image representation with the dimension of 257×768. Our voxel encoder consists sequentially of MLPs and residual networks while the voxel decoder is just the opposite. We trained the fMRI autoencoder end-to-end for 300 epochs, using AdamW (Loshchilov, 2017) with a cycle learning rate schedule starting from 3e-4. For the diffusion estimator, we set the timesteps $\mathcal{T}$ to 100, adopting a cosine noise schedule and 0.2 conditions drop. The diffusion network contained 6 Transformer blocks. Each block computes attention using 257 image tokens, 257 noised fMRI tokens, and 1 time embedding. We train it for 150 epochs using gradient clipping, with the same learning rate as our autoencoder. For hyperparameter $\beta$, we randomly sample from $U(0, 1)$. All components of our *MindSimulator* can be trained using a single NVIDIA Tesla V100 GPU. For a single subject, the training is about 12 GPU hours for the fMRI autoencoder and 20 GPU hours for the diffusion estimator. During the inference phase, it takes only 3̃00ms to synthesize an fMRI. Refer to Appendix A.1 for more implementation details.

### 4.3 EVALUATION METRICS

How do we evaluate the performance of fMRI encoding models? Previous studies (Gu et al., 2022; Wang et al., 2023; Luo et al., 2024) used voxel-level metrics, such as Pearson correlation, voxel-wise mean square error (MSE), and R-squared. However, local-focused voxel-level metrics are limited because they overlook global accuracy. Specifically, these metrics fail to evaluate whether synthetic fMRI accurately preserves the original neural response patterns, which are the root of human visual phenomena. We utilize images for an analogous explanation. As shown in Figure 2, we generate two predictions for a

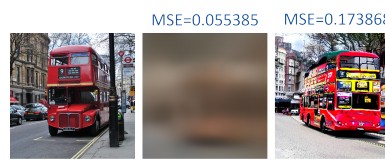

Figure 2: Analogical explanation for limitation of voxel-level metrics. The better low-level performance does not indicate a more accurate synthesis.

ground truth image. It can be seen that Prediction 1 has better pixel-level accuracy (pixel-wise MSE) but lacks recognizable global pixel patterns (image semantics), whereas Prediction 2 shows the opposite. Consequently, we might conclude that Prediction 2 is superior. Similarly, the same issues would arise with fMRI synthesis, highlighting the need to introduce semantic-level metrics.

Thanks to advanced research on fMRI visual decoding tasks (Scotti et al., 2024a;b), we can incorporate semantic-level evaluation for synthetic fMRI. The trained fMRI decoding model can interpret

individual neural response patterns and reconstruct original visual stimuli. Thus, when the synthetic fMRI aligns with the generalization capacity of the trained visual decoding model, if it retains the neural response pattern, it can be recognized by fMRI decoding models and the reconstructed visual stimuli should be similar to the seen visual stimuli. Based on the above reflections, we propose the semantic-level evaluation pipeline for synthetic fMRI, as shown in Figure 3. We utilize a decoding model to reconstruct visual stimuli based on synthetic fMRI and then compare it with ground truth stimuli using image reconstruction metrics. Among these metrics, PixCorr, SSIM, Alex(2), and Alex(5) are used to evaluate whether the synthetic fMRI retains perceptual-relevant neural patterns, while Incep, CLIP, Eff, and SwAV are used to evaluate concept-relevant neural patterns. As for the decoding model, we use the trained MindEye2 high-level model (Scotti et al., 2024b). More details on semantic-level metrics can be found in Appendix A.2.

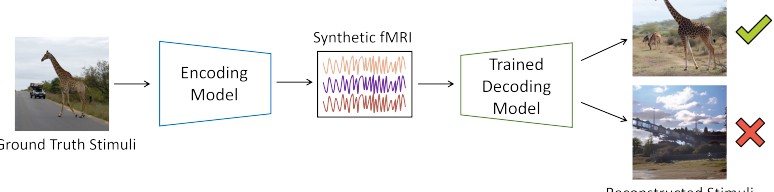

Figure 3: The proposed semantic-level evaluation pipeline for synthetic fMRI. We use trained visual decoding models to extract the semantics contained in the synthetic fMRI and compare them with the ground truth.

## 5 RESULTS

### 5.1 EVALUATION FOR SYNTHETIC FMRI

Generating accurate fMRI is crucial for localizing concept-selective regions. We evaluate the encoding accuracy of synthesized fMRI using the voxel-level and semantic-level metrics. For comparison, we select two representative encoding models as baselines: the linear regression model, known for its strong interpretability with wide applications in neuroscience research (Huth et al., 2016; Wang et al., 2023; Tang et al., 2023; Luo et al., 2024), and the Transformer encoding model (Adeli et al., 2023), which has performed exceptionally well in a fMRI encoding challenge (Gifford et al., 2023) and is increasingly used in recent studies (Liang et al., 2024; Beliy et al., 2024). Details on the implementation and training for the baseline encoding models can be found in Appendix A.3. We also compare the semantic-level metrics calculated using ground truth fMRI, serving as an upper bound of the encoding performance. The quantitative evaluation results are shown in Table 1.

Overall, our *MindSimulator* outperforms baselines in encoding accuracy. Additionally, its performance approaches the upper bound, suggesting minor differences in voxel-wise similarity and global neural response patterns between synthetic fMRI and ground truth fMRI. We further validated the above conclusion through visualization results. As shown in Figure 4, the fMRI synthesized by *MindSimulator* is significantly better than that of the linear regression model, indicating it contains more accurate neural response patterns, and thereby can be recognized by trained fMRI decoding models. Moreover, compared with seen visual stimuli, the differences in semantics are almost negligible. Such results are encouraging because they are sufficiently accurate and thereby allow us to explore neuroscience findings using synthetic fMRI instead of scarce ground truth fMRI.

In addition, two facts need to be further discussed. One is the decoded results from the synthesized fMRI have noise, which we attribute to minor voxel-wise differences. Since the decoding model is trained using ground truth fMRI, they struggle to accurately decode the mildly distorted synthetic fMRI. The other is that when we apply the multi-trial enhancement strategy, both MSE and semantic-level metrics improve simultaneously. This is out of our expectation, as increased voxel-wise reproducibility typically dilutes the accuracy of global neural response patterns. We attribute the observed fact to our correlated noise, which reduces variance in the neural pattern across multiple fMRI encoding trials, allowing multi-trial averaging to enhance semantic-level metrics.

### 5.2 OUT-OF-DISTRIBUTION GENERALIZATION

Further evaluation of the generalization performance of *MindStimulator* is essential. Although it has shown promising encoding performance on MSCOCO, neuroscience researchers may require its ap-

| Method | Voxel-Level | | Semantic-Level | | | | | | | |
|---|---|---|---|---|---|---|---|---|---|---|
| | Pearson↑ | MSE↓ | PixCorr↑ | SSIM↑ | Alex(2)↑ | Alex(5)↑ | Incep↑ | CLIP↑ | Eff↓ | SwAV↓ |
| GT fMRI (upper bound) | - | - | 0.278 | 0.328 | 95.2% | 99.0% | 96.4% | 94.5% | 0.622 | 0.343 |
| Linear Regressive | 0.334 | 0.394 | 0.174 | 0.266 | 85.4% | 94.2% | 90.1% | 87.2% | 0.728 | 0.432 |
| Transformer Encoding | 0.337 | 0.387 | 0.166 | 0.286 | 83.5% | 93.0% | 89.8% | 85.5% | 0.759 | 0.440 |
| MindSimulator (Trials=1) | 0.345 | 0.404 | 0.194 | 0.296 | 89.0% | 96.2% | 92.3% | 90.3% | 0.702 | 0.399 |
| MindSimulator (Trials=5) | **0.355** | **0.385** | **0.201** | **0.298** | **89.6%** | **96.8%** | **93.2%** | **91.2%** | **0.688** | **0.393** |

Table 1: Evaluation Results of fMRI synthesis accuracy. We report the average values for the 4 subjects. Our *MindSimulator* achieves optimal performance in both voxel-level metrics and semantic-level metrics.

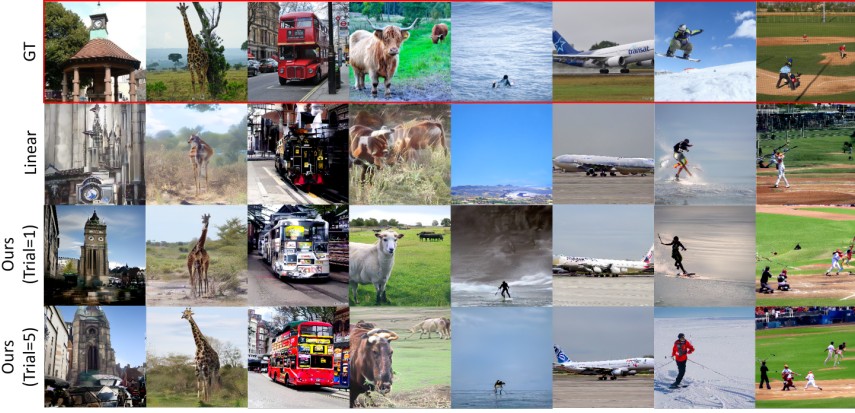

Figure 4: Visualization comparison between linear regression encoding and our *MindStimulator*. GT = seen visual stimuli. Linear = reconstruction from linear encoded fMRI. Ours = reconstruction from our encoding. The fMRI synthesized by our method has more accurate concepts, colors, backgrounds, and number of objects. We show the results of Subj01 and more results can be found in Appendix B. Zoom in for better viewing.

plication on other image datasets. Thus, its ability to encode images from different datasets is equally important. To evaluate this, we utilized CIFAR-10 and CIFAR-100 (Krizhevsky et al., 2009), which have distinct image distributions compared to MSCOCO. We select all 10,000 images from their test sets for encoding. Due to the lack of ground truth fMRI, we only compute the semantic-level metrics. The quantitative results are presented in Table 2, while qualitative visualizations are shown in Figure 5. We can see that encoding with CIFAR-10/100 images demonstrates only a minimal decrease in most metrics, and in some metrics, even shows improvement compared to MSCOCO. Additionally, the visualization results also indicate that the synthetic fMRI still retains accurate visual semantics. The results demonstrate that *MindStimulator* has out-of-distribution generalization capability, suggesting we can use a wider range of image data for neuroscience exploration.

| Datasets | Semantic-Level | | | | | | | |
|---|---|---|---|---|---|---|---|---|
| | PixCorr↑ | SSIM↑ | Alex(2)↑ | Alex(5)↑ | Incep↑ | CLIP↑ | Eff↓ | SwAV↓ |
| MSCOCO | 0.201 | 0.302 | **89.5%** | **96.8%** | **91.5%** | 88.7% | **0.724** | **0.409** |
| CIFAR-10 | **0.269** | 0.406 | 88.5% | 94.3% | 84.3% | **90.5%** | 0.898 | 0.645 |
| CIFAR-100 | 0.260 | **0.420** | 86.6% | 93.0% | 82.8% | 86.3% | 0.916 | 0.659 |

Table 2: Evaluation of out-of-distribution generalization. We report the results of Subj01. Our *MindSimulator* demonstrates excellent fMRI synthesis performance on out-of-distribution datasets.

## 5.3 ABLATION

We conduct ablation experiments to investigate the necessity of each design in *MindSimulator*. First, we ablate the Voxel Decoder, aligning outputs of the Voxel Encoder with the image latent space under the supervision of SoftCLIP, while the diffusion estimator directly predicts the voxels instead of fMRI representations. Second, we ablate the fMRI-stimuli joint latent space, training the entire fMRI Autoencoder solely using the MSE loss. Furthermore, we ablated the high-dimensional fMRI representation space by removing the fMRI Autoencoder, which means that the diffusion estimator is trained only in the original voxel space. Finally, we also ablate the correlated noise. The experimental results are displayed in Table 3. As we can see, each component of our *MindSimulator* plays a role in improving the fMRI encoding performance. In addition, they also contribute to the stable convergence of our diffusion estimator.

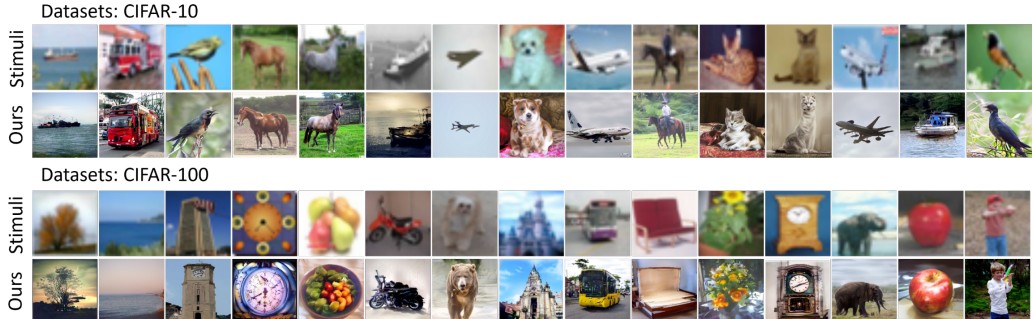

Figure 5: Comparison between CIFAR-10/100 images (Stimuli) and corresponding reconstructing results from MindSimulator's synthetic fMRI (Ours). The original stimuli are upsampling to 224×224.

| Methods | Voxel-Level | | Semantic-Level | | | | | | | |
|---|---|---|---|---|---|---|---|---|---|---|
| | Pearson↑ | MSE↓ | PixCorr↑ | SSIM↑ | Alex(2)↑ | Alex(5)↑ | Incep↑ | CLIP↑ | Eff↓ | SwAV↓ |
| MindSimulator | 0.322 | 0.418 | 0.204 | 0.302 | 90.9% | 96.5% | 93.0% | 89.8% | 0.712 | 0.407 |
| -w/o Voxel Decoder | 0.287 | 0.457 | 0.196 | 0.289 | 89.8% | 95.8% | 87.7% | 86.6% | 0.755 | 0.428 |
| -w/o Joint Latent Space | 0.215 | 0.524 | 0.181 | 0.293 | 87.3% | 93.5% | 85.3% | 82.5% | 0.791 | 0.456 |
| -w/o fMRI Autoencoder | 0.203 | 0.843 | 0.152 | 0.295 | 82.8% | 89.4% | 78.2% | 75.2% | 0.855 | 0.506 |
| -w/o Correlated Noise | 0.316 | 0.431 | 0.200 | 0.302 | 89.8% | 96.2% | 91.5% | 89.6% | 0.726 | 0.404 |

Table 3: Ablation experiments on each component of *MindSimulator*. We report the results of Subj01. Each component of our method is crucial for improving fMRI encoding performance.

# 6 LOCALIZING CONCEPT-SELECTIVE REGIONS

Our evaluations have demonstrated that synthetic fMRI exhibits neural response patterns that resemble ground truth fMRI. This allows us to synthesize massive fMRI data for neuroscience research, especially in the localization of concept-selective regions. Our localization exploration involves two steps. First, we select concept-oriented visual stimuli and synthesize the corresponding fMRI. We then apply the same statistical analyses as fLoc (Stigliani et al., 2015) to identify concept-selective regions, validating these results against established empirical findings. In the second step, we predict novel concept-selective regions and confirm these predictions through voxel ablation experiments.

## 6.1 PREDICT EMPIRICAL REGIONS

The NSD dataset localized places-, bodies-, faces-, and words-selective regions through functional localizer (fLoc) experiments. As illustrated in Figure 6, simple observation can reveal great overlap in the regions associated with fLoc words-selective and fLoc faces-selective, while fLoc place-selective and fLoc bodies-selective do not. Consequently, we intend to predict places- and bodies-selective regions using fMRI synthesized by our *MindSimulator*, and evaluate the predictions using empirical findings from the existing NSD fLoc experiments.

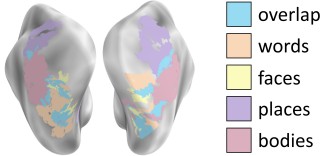

Figure 6: The empirical findings of faces-, bodies-, places-, and words-selective regions in NSD fLoc.

The first step in localization involves selecting concept-oriented visual stimuli. To achieve this, we utilize the pre-trained CLIP model for zero-shot classification, which compulsory assigns MSCOCO images to the target concept categories. After these, we select the top-k images with the highest classification probability as visual stimuli used for further exploration. In the NSD fLoc experiments, hundreds of visual stimuli are used, so we similarly select between 100 and 1000 top-ranking images. Figure 7 presents a subset of the top 100 selected images and compares them with stimuli used in fLoc experiments. Significant distribution differences exist.

We encode the selected images into fMRI voxels using both *MindSimulator* and the commonly adopted linear regression as encoding models. Based on the synthetic fMRI, we compute voxel-wise statistical significance by performing a one-sample t-test and Bonferroni correction. This allows us to identify concept-selective regions by setting a significance threshold. To evaluate the accuracy of our data-driven region localization, we use Accuracy and F1-score as metrics, comparing the predicted regions with empirical findings. Additionally, we compute the semantic specificity of the images used for encoding, which reflects the strength of relevance between the selected images

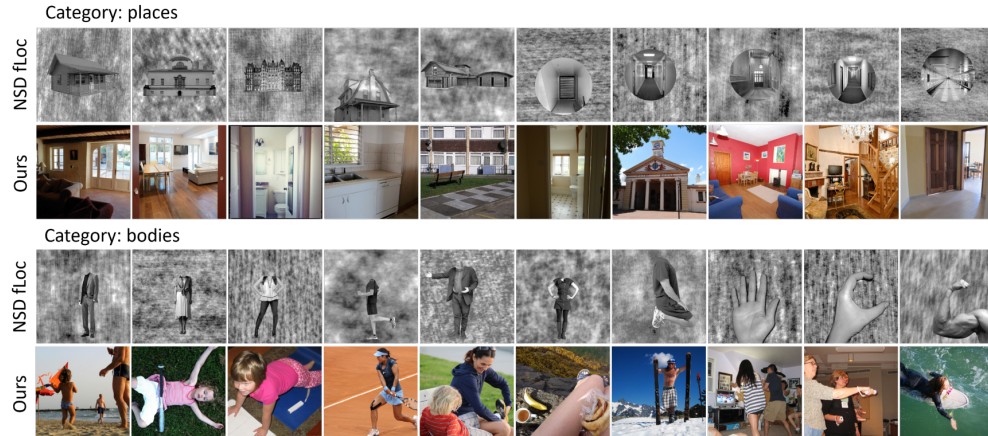

Figure 7: Comparison between visual stimuli used in NSD fLoc experiments and images we use for localizing concept-selective regions. The fLoc experiments use visual stimuli that place targets in unnatural scenes, whereas we use the target concept in real scenes. They exhibit huge image distribution differences.

| # Images | places-Specificity | places-Acc↑ | | places-F1↑ | | bodies-Specificity | bodies-Acc↑ | | bodies-F1↑ | |
|---|---|---|---|---|---|---|---|---|---|---|
| | | Linear | Ours | Linear | Ours | | Linear | Ours | Linear | Ours |
| Top 100 | 0.9608 | 32.0% | **52.6%** | 0.461 | **0.573** | 0.9988 | 45.8% | **91.7%** | 0.570 | **0.674** |
| Top 200 | 0.9391 | 30.0% | **45.6%** | 0.444 | **0.563** | 0.9977 | 42.1% | **84.2%** | 0.551 | **0.727** |
| Top 300 | 0.9189 | 29.2% | **41.3%** | 0.437 | **0.540** | 0.9968 | 40.7% | **81.4%** | 0.543 | **0.740** |
| Top 500 | 0.8834 | 28.5% | **37.3%** | 0.430 | **0.513** | 0.9953 | 39.2% | **75.3%** | 0.532 | **0.732** |
| Top 1000 | 0.8084 | 27.9% | **33.5%** | 0.425 | **0.483** | 0.9918 | 37.9% | **64.2%** | 0.523 | **0.693** |

Table 4: Localization evaluation of places- and bodies-selective regions. We report the results of Subj01. The results are average values obtained with 3 different random seeds
.

and the target concept. This specificity is calculated by averaging the classification probabilities of all images. More implementation details on localization can be found in Appendix A.4, and the evaluation results are presented in Table 4.

The results show that the localization accuracy using fMRI synthesized by *MindSimulator* is satisfactory, particularly for bodies-selective regions, despite notable image distribution differences between the NSD fLoc experiments and our selected images. This highlights the feasibility of our data-driven approach for localizing concept-selective regions. Moreover, our localization for places- and bodies-selective regions outperforms the linear regression encoding model in both Accuracy and F1 metrics, which we attribute to *MindSimulator* can synthesize higher-quality fMRI. We also observe that as the number of selected images increases, semantic specificity tends to decrease. In addition, there is a positive correlation between localization accuracy and semantic specificity, indicating that enhancing accuracy requires a focus on encoding images with higher semantic specificity.

## 6.2 EXPLORING NOVEL REGIONS

After confirming that the synthetic fMRI can effectively localize concept-selective regions, we explore the localization of novel regions. Statistically, we can pinpoint any region associated with a given concept if we select hundreds of images with high semantic specificity from image datasets like MSCOCO or CIFAR-10/100. As a pioneering exploration, we focus on several concepts of interest, including surfer-, plane-, food-, and bed-selective regions. We employ CLIP zero-shot classification to identify the top 200 images with the highest semantic specificity from MSCOCO and then synthesize fMRI. Subsequently, we perform statistical tests to evaluate voxel-wise activation significance and conduct region localization using the same thresholds. Figure 8 illustrates part of the selected images and their corresponding localization results.

The localized concept-selective regions are predominantly located in the higher visual cortex, as delineated by NSD. This aligns with prior neuroscience knowledge, which indicates that the lower visual cortex shows selectivity for colors and shapes, while the higher visual cortex is selective for specific concepts. In addition, it can be seen that the regions localized by different concepts are scattered across the cortex with minimal overlap. This suggests that our approach could be extended to create a concept atlas of the human brain. We also find that our localization often consists of

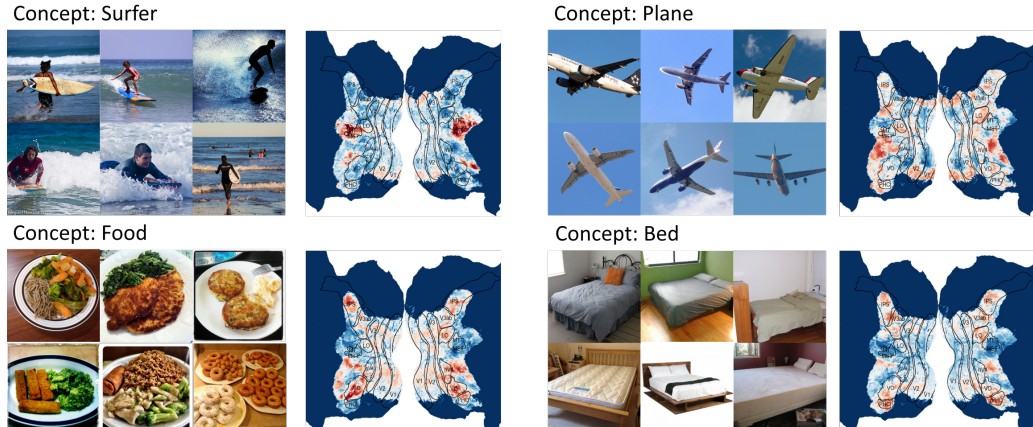

Figure 8: Localized concept-selective regions according to synthetic fMRI. The red represents the selected regions and darker colors reflect higher confidence. We show the results for Subj01. The results of other subjects and inter-subject comparisons can be found in Appendix B. Zoom in for better viewing.

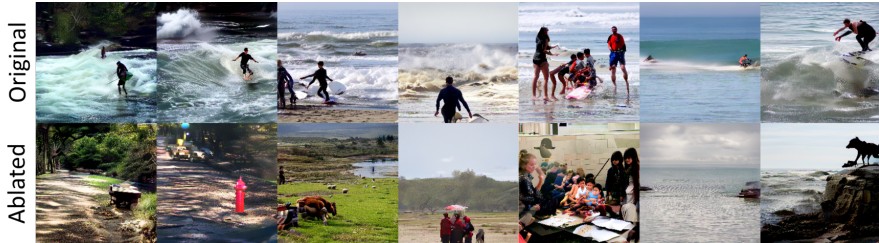

Figure 9: Results of voxel ablation experiments. We mask the surfer-selective region. More precisely, we set the corresponding 1017 voxels (around 6.5%) to negative values. Original = Reconstruction from synthetic fMRI. Ablated = Reconstruction from masked synthetic fMRI. It can be observed that when the concept region is masked, the two concepts of "surfer man" and "sea" are dissolved separately.

several disconnected regions, which we attribute to the coupling of related concepts. For example, the concept of "surfer" is often coupled with the concepts of "sea" and "human".

To verify the accuracy of the localized novel concept regions, we conduct voxel ablation. We mask the synthetic fMRI based on the localized concept-selective regions and then decode it using trained decoding models. To provide a more comprehensive presentation of ablation results, we take "surfer" for illustration. As shown in Figure 9, when the regions are masked, the reconstructed images lose corresponding concept objects, providing partial validation for our localization.

## 7  CONCLUSION

In this paper, we introduce MindSimulator, a generative fMRI encoding model that utilizes a diffusion model to learn the fMRI distribution conditional on given visual stimuli in a high-dimensional fMRI-stimuli joint latent space. By employing multi-trial enhancement for sampling, we synthesize massive fMRI data. Our experiments demonstrate that MindSimulator's synthetic fMRI outperforms existing regressive encoding models across both voxel-level and semantic-level metrics, with strong generalization on common image datasets. Leveraging synthetic fMRI, we conduct data-driven neuroscience explorations, localizing wide-studied concept-selective regions and validating these results against empirical findings. We believe that our approach of utilizing synthetic data to enlarge scarce fMRI datasets and then conduct neuroscience research offers an alternative complement to traditional approaches and provides novel hypothetical priors for future exploration.

## ACKNOWLEDGEMENTS

This research is supported by the National Natural Science Foundation of China (No. 62376198), the National Key Research and Development Program of China (No. 2022YFB3104700), Shanghai Baiyulan Pujiang Project (No. 08002360429).

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

# A  ADDITIONAL IMPLEMENTATION DETAILS

## A.1  ADDITIONAL IMPLEMENTATION DETAILS ON MINDSIMULATOR

**fMRI Autoencoder.** The autoencoder consists of a voxel encoder and a voxel decoder. The voxel encoder first maps fMRI (12682-15724 voxels, varies across subjects) into a 256-dimensional latent space using ridge regression. It then further processes the embedding in the latent space using 2 blocks, each containing multiple Linear layers, LayernNorm layers, activation layers, and residual connections. Finally, a linear projector projects the 256-dimensional latent embedding to the high-dimensional fMRI representation of $257 \times 768$. And the voxel decoder arranges the three components exactly in reverse. The fMRI autoencoder is trained end-to-end under the supervision of MSE loss and SoftCLIP loss, with both losses weighted equally at 1.

We also conduct an ablation study on the fMRI autoencoder, investigating the effects of latent space dimensions, the number of residual blocks, and whether end-to-end training is performed (if not end-to-end, we first use SoftCLIP supervision to learn the joint latent space and then train the voxel decoder). The ablation results are shown in Table 5, where we evaluate the MSE between ground truth voxel and encoding-then-decoding voxel (voxel input) as well as the MSE between ground truth voxel and voxel decoded form image representations (image input). We find that increasing the latent space dimensions and the number of residual blocks does not necessarily lead to performance improvements, so we ultimately opt for a smaller configuration to reduce memory overhead. Additionally, we discover that end-to-end training results in a lower MSE of voxel input. Furthermore, the extreme MSE metrics of image input indicate that fMRI representation and image representation are disjointed, underscoring the necessity of introducing a diffusion estimator.

| Hidden dim | # blocks | End-to-End | MSE↓ | |
| --- | --- | --- | --- | --- |
| | | | voxel input | image input |
| 256 | 2 | ✓ | 0.227 | 7.470 |
| 256 | 4 | ✓ | 0.240 | 8.305 |
| 512 | 2 | ✓ | 0.252 | 5.295 |
| 256 | 2 | ✗ | 0.401 | 1.687 |
| 256 | 4 | ✗ | 0.411 | 2.405 |
| 512 | 2 | ✗ | 0.343 | 8.124 |

Table 5: Ablation Results for fMRI Autoencoder.

**Diffusion Estimator.** The diffusion estimator contains 6 Transformer blocks. The inputs are 257 image tokens, 257 noised fMRI tokens, and 1 time embedding. Its output is 257 denoised fMRI tokens. We add absolute positional embeddings to the noised fMRI tokens and do not use learnable query tokens, because this significantly saves on memory. As for attention, we simply use bidirectional attention instead of causal attention.

## A.2  ADDITIONAL DETAILS ON EVALUATION METRICS

**Decoding Model.** We use the trained MindEye2 (Scotti et al., 2024b) as our decoding model for recognizing the visual semantics contained in synthetic fMRI. At this stage, all decoding models have adopted a similar approach. They first utilize contrastive learning to align fMRI voxels with the image latent space, then transform the fMRI representation into the image representation. Finally, the transformed fMRI representation is used as the text condition input for trained T2I models (Rombach et al., 2022; Xu et al., 2023) to reconstruct the image. Obviously, due to encoding as the reverse process of decoding, our MindSimulator has been greatly inspired by those decoding models.

**Image Reconstruction Metrics.** PixCorr denotes the pixel-wise correlation between ground truth image and reconstruction results. SSIM denotes Structural similarity index metric (Wang et al., 2004) between ground truth and reconstructions. Alex(2), Alex(5), Incep, and CLIP are metrics that refer to two-way identification (chance = 50%) using different models. Alex(2) denotes two-way comparisons are performed with the second layer of AlexNet (Krizhevsky et al., 2012), Alex(5) denotes comparisons with the fifth layer of AlexNet, Incep denotes comparisons with the last pooling layer of InceptionV3 (Szegedy et al., 2016), and CLIP denotes comparisons with the final layer of CLIP ViT-L/14 (Radford et al., 2021). Two-way identification refers to percent correct across comparisons gauging if the original image embedding is more similar to its paired voxel embedding

or a randomly selected voxel embedding. Eff and SwAV refer to the average correlation distance with EfficientNet-B1 (Tan & Quoc) and SwAV-ResNet50 (Caron et al., 2020). The average time cost for evaluating a single fMRI is within 1-2 seconds.

## A.3 DETAILS ON BASELINES

**Linear Regression Encoding.** The linear regression encoding model consists of two linear layers, the first of which projects the $257 \times 768$ image representation directly into the 2048 latent space, and the second of which projects the latent into the voxel dimension (such as 15724). We train this linear model for 150 epochs using AdamW, and the learning rate decreases linearly from 3e-4. For inference, we use the checkpoint with the smallest MSE metric in the validation set.

**Transformer Encoding.** The Transformer Encoding model uses 10 ROI queries. Each ROI query is a learnable token, corresponding to a ROI. We consider a total of 10 ROIs, including lh-faces, lh-bodies, lh-places, lh-words, lh-remains, rh-faces, rh-bodies, rh-places, rh-words, and rh-remains. The 10 ROI queries and image tokens are input to a single-layer Transformer with one cross-attention and one self-attention operation. Each output ROI query is then mapped using a single linear layer to fMRI voxel. The predicted voxel is then multiplied by a mask that is zero everywhere except for the vertices belonging to that ROI, ensuring that the gradient feeding back from the loss will only train linear mappings to the voxel of the queried ROI. During inference, the predicted voxel from different ROI readouts will then be combined using the same masks to generate the synthetic fMRI. We train this encoding model for 150 epochs using AdamW, and the learning rate decreases linearly from 3e-4. Finally, the checkpoint with the smallest MSE metric is used.

## A.4 ADDITIONAL DETAILS ON LOCALIZATION

**Images.** We used the MSCOCO images adopted in the NSD experiment (a total of 73,000 images, including 9,000 images unique to each subject and 1,000 images shared among subjects). Note that each subject had fMRI data corresponding to only 10,000 images.

**Prompts.** In the NSD fLoc experiments, researchers select visual stimuli from fixed categories. Specifically, places-stimuli contain "house" and "hallway", bodies-stimuli contain human "body" and "limb", faces-stimuli contain real "adult face" and "children face", and words-stimuli contain "characters" and "numbers". Therefore, to validate our localization with places-, bodies, faces- and words-selective regions, we utilize the following prompts for zero-shot classification: [`"houses or corridors"`, `"human bodies or human limbs"`, `"real human faces"`, `"words or numbers"`]. As for the exploration of novel concept-selective regions, we use the most common prompts, [`"a photo of surfer"`, `"a photo of plane"`, `"a photo of food"`, `"a photo of bed"`].

**Significance Thresholds.** In the NSD fLoc experiments, the significance threshold is set to 0, which is very generous. In this setting, the range of concept-selective regions is very large, for example, Subj06's bodies-selective region even contains 4887 voxels. In our concept-selective voxel localization, the significance factor of the t-test is set to 0.01. In this setting, the selected region usually contains voxels ranging from hundreds to a few thousand.

# B ADDITIONAL RESULTS

## B.1 ADDITIONAL METRICS FOR SYNTHETIC FMRI

We first include R-squared $R^2$, which is commonly used in neuroscience studies. The results are presented in Figure 10. We also evaluated the synthetic fMRIs with additional metrics covering the similarity of functional connectivity, spatial structure similarity, and the similarity of fMRI Representations. The evaluation metrics are described as follows.

**Similarity of Functional Connectivity Graphs**. We use the functional areas provided by NSD. Dimensionality reduction is performed on the voxels within each ROI using PCA to ensure they have the same length. Subsequently, Pearson correlation was calculated among the different regions to obtain the functional connectivity maps. The similarity between the functional connectivity maps of the synthetic and real fMRI data is then evaluated using the mean absolute error.

**Spatial Gradients**. We compute the spatial gradients of each sample in the test set across all directions and then calculate the absolute difference between the ground-truth fMRI and the synthetic fMRI. The average results of each voxel are reported.

**Similarity of fMRI Representations**. we compute the two-way classification accuracy of fMRI representations.

All results are reported in Table 6. It can be observed that the results of different methods are similar.

| Methods | Connectivity Graphs↓ | Spatial Gradients↓ | fMRI Rep. Similarity↑ |
|---|---|---|---|
| Linear Encoding Model | 0.127466 | **0.536810** | 99.27% |
| Transformer Encoding Model | 0.123541 | 0.543872 | 99.43% |
| MindSimulator | **0.122895** | 0.538561 | **99.77%** |

Table 6: Additional metrics for synthetic fMRI.

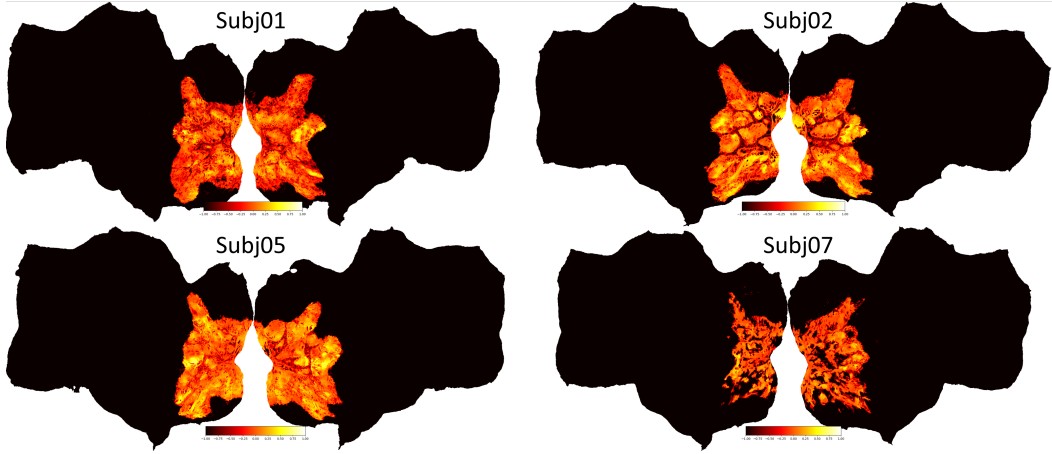

Figure 10: The $R^2$ metric of synthetic fMRI for each subject.

## B.2 ADDITIONAL VISUALIZATION OF SYNTHETIC FMRI

In addition, we further provide the reconstructed image from the synthetic fMRI for 4 subjects, as shown in Figure 11 to Figure 14.

## B.3 ADDITIONAL RESULTS ON EXPLORING NOVEL REGIONS

We display the cortical selective regions of the same concepts among different subjects. As illustrated in Figure 16 to Figure 18, despite subtle differences, the activation of the same concepts in different subjects revealed similar patterns, with the strongest areas of activation typically occurring in the same locations. This indicates a high degree of similarity among individual brains.

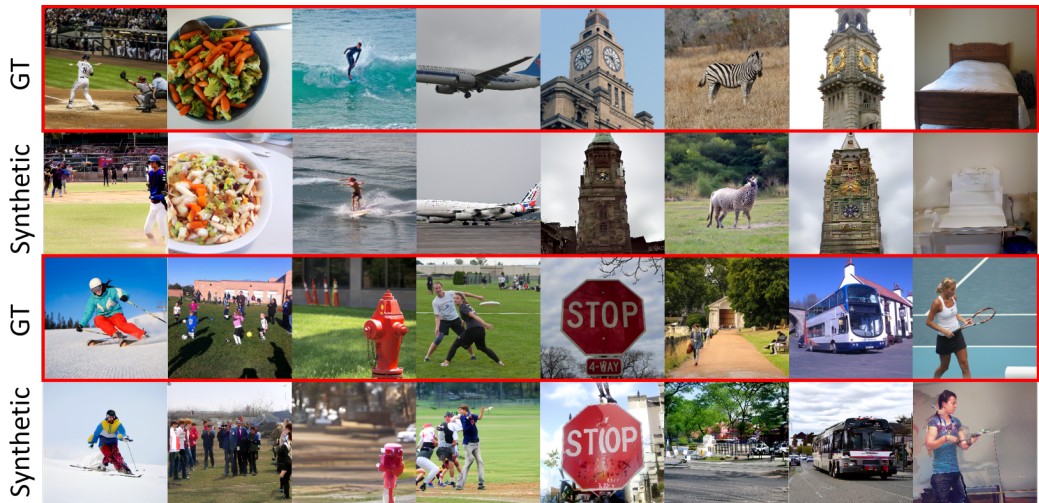

Figure 11: Additional visualization of reconstructed images from Subj01 synthetic fMRI. GT = seen visual stimuli. Synthetic = reconstructed images from synthetic fMRI. Randomly selected.

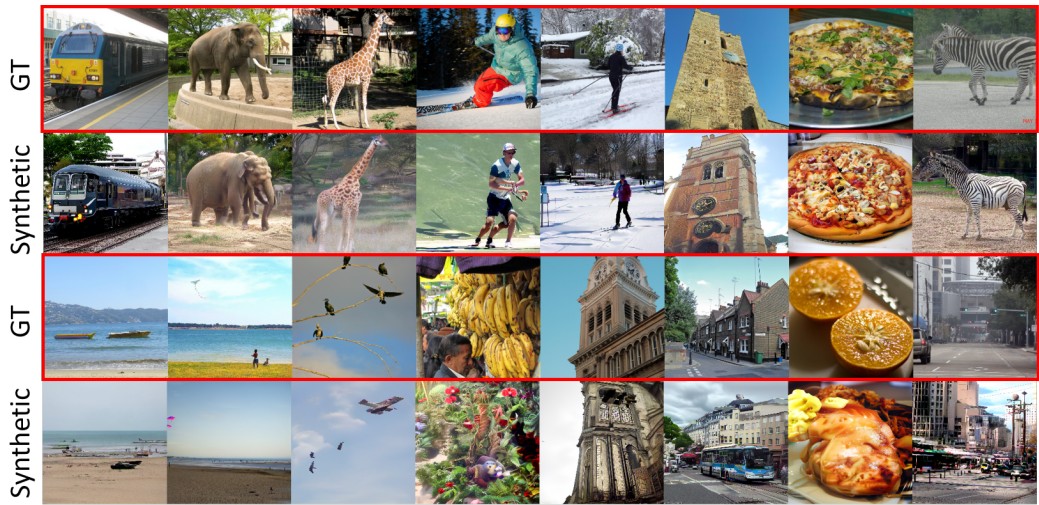

Figure 12: Additional visualization of reconstructed images from Subj02 synthetic fMRI. GT = seen visual stimuli. Synthetic = reconstructed images from synthetic fMRI. Randomly selected.

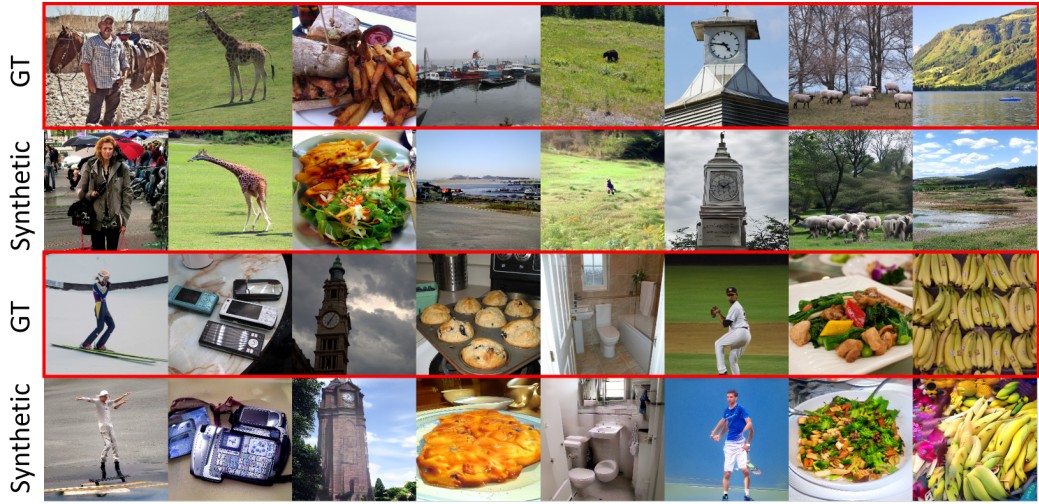

Figure 13: Additional visualization of reconstructed images from Subj05 synthetic fMRI. GT = seen visual stimuli. Synthetic = reconstructed images from synthetic fMRI. Randomly selected.

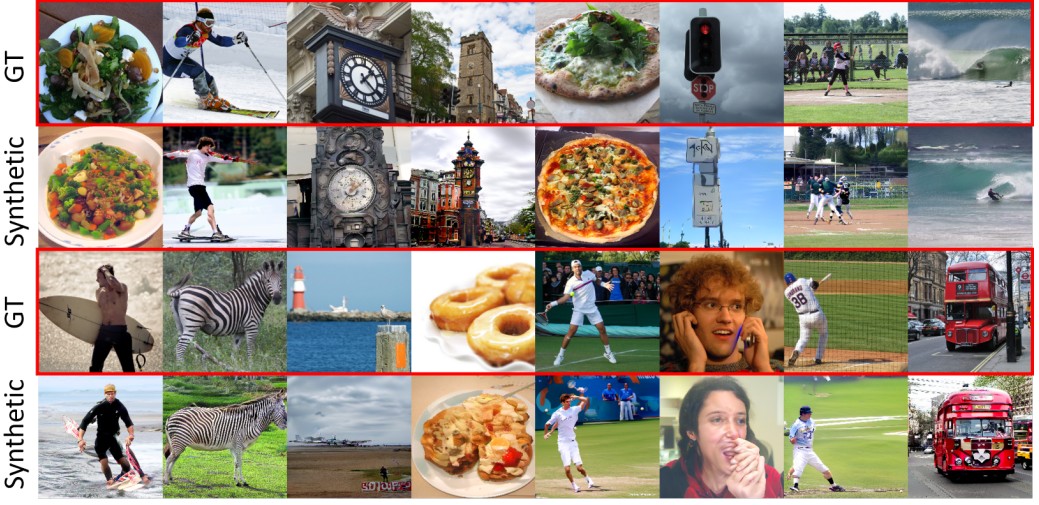

Figure 14: Additional visualization of reconstructed images from Subj07 synthetic fMRI. GT = seen visual stimuli. Synthetic = reconstructed images from synthetic fMRI. Randomly selected.

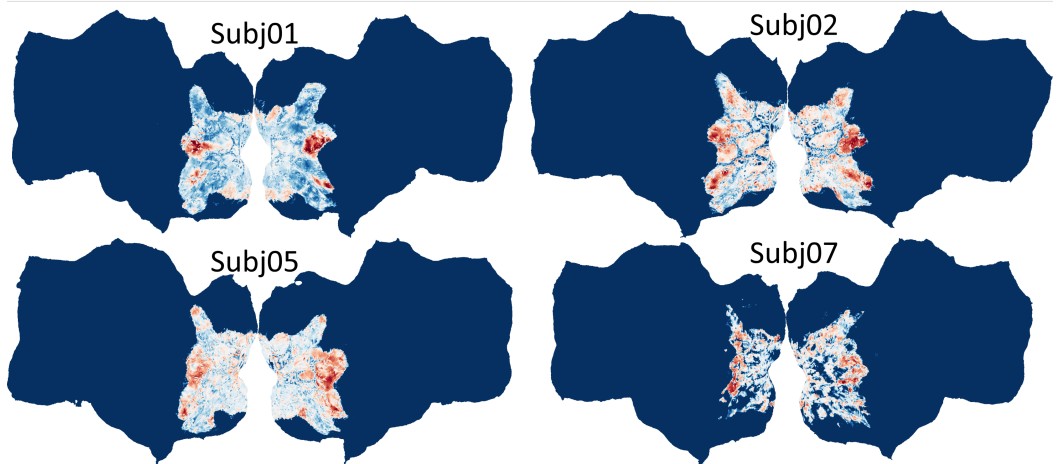

Figure 15: Surfer-selective regions of each subject.

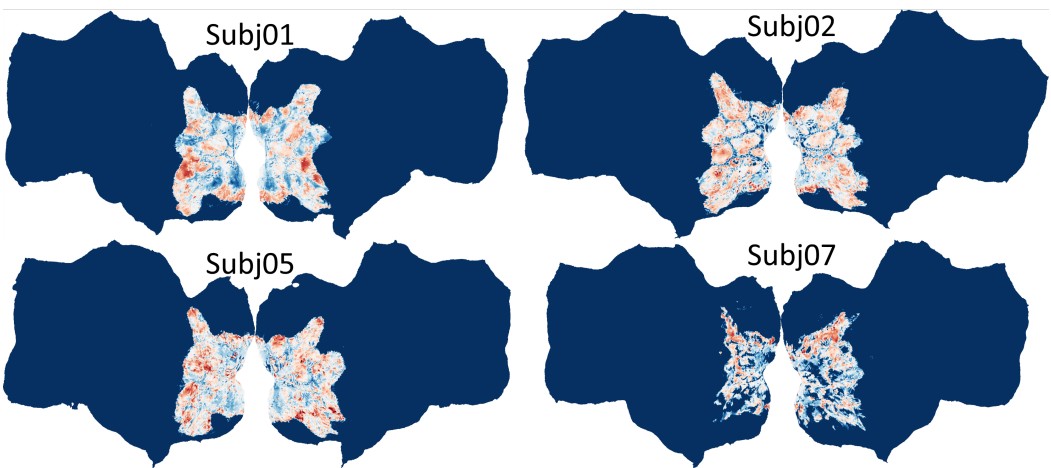

Figure 16: Plane-selective regions of each subject.

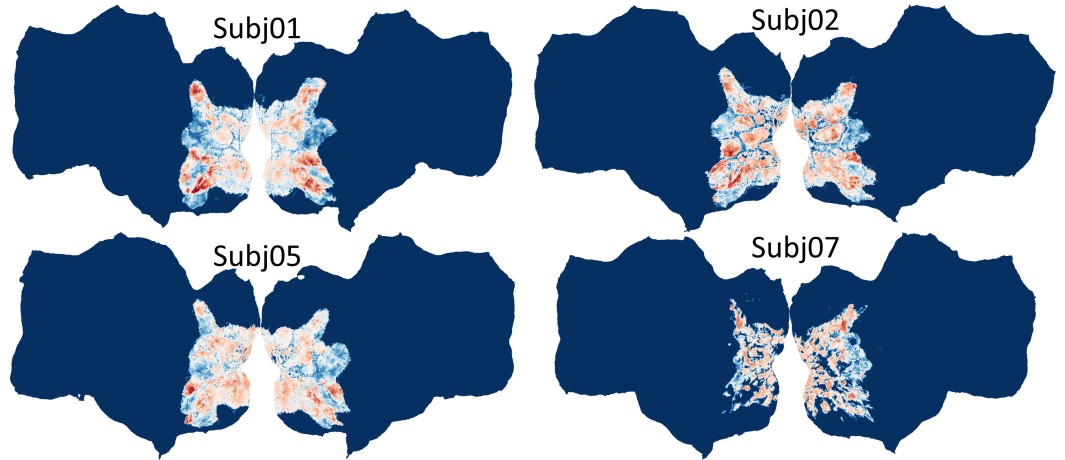

Figure 17: Food-selective regions of each subject.

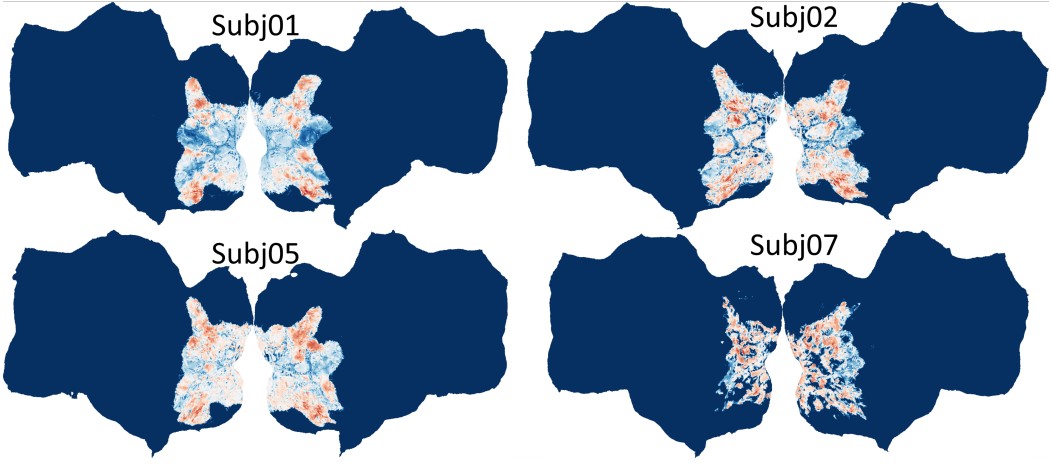

Figure 18: Bed-selective regions of each subject.

## C    VALIDITY OF SEMANTIC METRICS

The question of whether semantic metrics can effectively identify the semantics contained in fMRI signals is one that requires further verification. We explore this through noise decoding experiments. We use three types of noise as the input of the trained decoding model (MindEye2). The results are shown in Table 7.

| Method | PixCorr↑ | SSIM↑ | Alex(2)↑ | Alex(5)↑ | Incep↑ | CLIP↑ | Eff↓ | SwAV↓ |
|---|---|---|---|---|---|---|---|---|
| Random Input | 0.027 | 0.178 | 50.7% | 50.0% | 50.8% | 50.1% | 0.980 | 0.632 |
| Shuffled GT fMRI | 0.023 | 0.143 | 50.4% | 51.0% | 50.4% | 50.2% | 0.983 | 0.642 |
| Shuffled Synthetic fMRI | 0.021 | 0.114 | 50.4% | 51.4% | 49.9% | 50.7% | 0.986 | 0.650 |
| MindSimulator | **0.201** | **0.298** | **89.6%** | **96.8%** | **93.2%** | **91.2%** | **0.688** | **0.393** |

Table 7: Validity experiments of semantic metrics.

It can be seen the decoding performance of shuffled fMRI is comparable to that of random input. Most performance of baselines reach their theoretical minimum values, such as Pearson correlation and pixel similarity (PixCorr) approaching 0; Alex(2), Alex(5), Incep, and CLIP nearing 50%. These results suggest that trained decoding models can successfully capture the visual semantics contained in fMRI rather than other irrelevant content. At the same time, this also indicates that reconstruction can only be achieved when the fMRI contains real visual semantics. We further display the reconstructed results of noise input. As shown in Figure 19, no clear visual semantics can be reconstructed from noise.

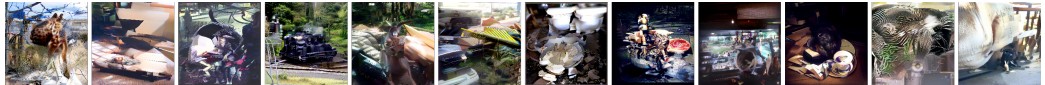

Figure 19: Reconstructed results from noise. No clear visual semantics are contained in the reconstructed images. Please zoom in for better viewing.

## D    COMPARISON OF DIFFERENT CONCEPT-SELECTIVE REGION LOCALIZATION METHODS

In concurrent papers, Shen et al. (2025) proposed a concept-selective region localization method based on Grad-CAM (Selvaraju et al., 2017). We briefly compare it with our method, and the corresponding results are shown in Figure 20.

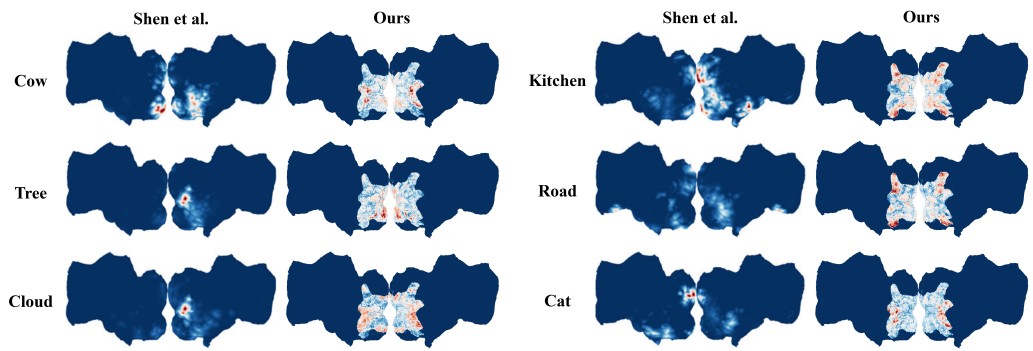

Figure 20: Comparison between Grad-CAM-based localization and our method. Our method localizes the concept-selective regions more focused on the higher-level visual cortex.

## ETHIC STATEMENT

Our research adheres to the ICLR Code of Ethics. All experiments in this paper are conducted using open-source datasets, and no potential ethical concerns are associated with this work.

## REPRODUCIBILITY STATEMENT

This study employs generative encoding models to synthesize fMRI data, facilitating the localization of concept-selective regions. All preprocessed data, code, and model parameters used in our research will be made publicly available upon publication. Detailed protocols for data preprocessing, model training, and evaluation have been provided in our manuscript, enabling independent reproduction.

## FUTURE WORK

In future work, we will validate the effectiveness of MindSimulator on more fMRI datasets and verify the validity of data-driven concept selection region localization through neuroscience experiments. We will also conduct more detailed exploration of the concept selection regions using synthetic fMRI, develop a human brain concept map, and study individual differences. Additionally, we are exploring the potential of encoding model feedback to improve cross-subject decoding performance using synthetic fMRI.

