# OpenReview forum: "MindSimulator: Exploring Brain Concept Localization via Synthetic fMRI"
_ICLR.cc/2025/Conference — ICLR 2025 Poster_

### Official Review · Reviewer_Atef · 2024-10-31

**Soundness:** 2
**Presentation:** 3
**Contribution:** 3
**Rating:** 6
**Confidence:** 4

**Summary:**

This paper proposes "MindSimulator", a framework that synthesizes fMRI data based on visual stimuli through an fMRI autoencoder, diffusion estimator, and inference sampler. The authors first assessed the performance of the fMRI autoencoder and diffusion estimator using various metrics, demonstrating their capability to generate high-quality fMRI data. They then used the synthesized fMRI data to explore correlations between manually selected images and brain activity, offering new insights for neuroscience research.

**Strengths:**

1. The paper offers a novel perspective by applying well-established fMRI visual decoding models for fMRI signal synthesis, with thorough validation to demonstrate reliability.
2. This study introduces a new tool for exploring *concept-selective regions*, significantly enhancing the flexibility of investigating how specific human visual representations of concepts are spatially distributed in the brain.

**Weaknesses:**

1. The algorithm for localizing concept-selective regions may lack sufficient validation, as the paper only compares this approach to fLoc, without further support from neuroscience literature. Consideration of alternative methods, like Neurosynth or Text2Brain, could strengthen the results, as these methods allow a broader selection of concepts correlated with brain activity, potentially detecting concepts not covered by fLoc.

2. In the *Evaluation Metrics* section, the method of validating generated fMRI data based on the quality of generated images may not be reliable due to its reliance on a separate trained decoding model. Given the complexity of visual decoding from fMRI, this dependence could reduce the robustness of the evaluation. Exploring alternative evaluation methods, such as comparing generated data with latent representations in the voxel encoder’s latent space, might provide more direct validation.

**Questions:**

1. In the *Inference Sampler* section, the mention of "resting-state brain activity fMRI" could be misleading, suggesting that the model can generate resting-state fMRI data. However, I could not find evidence of any relevant dataset being used. Could the authors clarify this point?
2. In the *Out-of-Distribution Generalization* section, CIFAR-10/100 was used, and metrics were calculated based on images decoded from synthesized fMRI data. As noted in the weaknesses, this approach may introduce bias. Why not use an image-fMRI dataset, like THING-fMRI, to compute metrics directly on fMRI data?

---

> ### Author Response · Authors · 2024-11-24
> **Author Rebuttal for Reviewer Atef**
>
> Dear Reviewer Atef,
>
> Thanks for your constructive comments and valuable advice. We address your concerns one by one as below.
>
> `W1:`
>
> We have carefully read the two papers [1, 2] you mentioned and find that they are **unsuitable** to be used as our baseline methods for the following reasons:
>
> + **The inconsistency of stimuli category**. In our study, we focus on the localization of visual concepts (elicited by visual stimuli). However, the data collected by Neurosynth primarily involve fMRI obtained under text-based stimuli or real-world physical stimuli. Such differences make the two completely incomparable.
>
> + **Subject differences**. A more significant issue lies in the difference in subjects. Note that the regions of interest (ROIs) vary between individuals. Our method and fLoc share the same subjects in the NSD dataset, whereas the localization from Neurosynth/Text2Brain is unrelated to the NSD subjects, making comparisons infeasible.
>
> `W2:`
>
> Note that evaluating the semantic accuracy of synthetic fMRI data poses significant challenges. As you mentioned, using generated images from trained decoding models (MindEye2) introduces complexity and strong dependencies. However, **our evaluation approach has two advantages**:
>
> + **Interpretability**: The generated images provide a more intuitive visualization of the visual semantic content derived from the fMRI data, facilitating further analysis and identifying potential deficiencies. In contrast, relying on fMRI latent representations reduces computation costs, but, falls short in intuitive visualization and explanations.
> + **High recognition and reproducibility**: The trained MindEye2 models are widely recognized for their ability to extract and visualize visual information with fMRI data, capably offering a more authoritative and reproducible evaluation compared to a specific custom-trained voxel encoder.
>
> Additionally, the method you proposed is also an efficient and lightweight strategy for fMRI semantic-level evaluation. **Following your suggestion, we use the two-way classification accuracy of fMRI latent representations for evaluation**. The results presented as follows show the best semantic accuracy of our MindSimulator. However, it can be observed that all values of these results are very high, resulting in insignificant differentiation and comparison. We have included these results in **Appendix B.2**.
>
> | Methods                    | Accuracy |
> | - | - |
> | Linear Encoding Model      | 99.27%   |
> | Transformer Encoding Model | 99.43%   |
> | MindSimulator (Trials=1)   | 99.67%   |
> | MindSimulator (Trials=5)   | 99.77%   |
>
> `Q1:`
>
> We must clarify that the resting-state fMRI is **NOT the output** of our Diffusion Estimator; in contrast, **it is used as the input of our model**. We replace pure Gaussian noise with the noised resting-state fMRI representation, which is designed to simulate the transition from the previous brain state (resting state) to the current brain state.
>
> The approach of utilizing resting-state brain activity fMRI as the inference initialization is introduced as **an exploration to incorporate the temporal dependencies of fMRI**. Recent research [3] in fMRI encoding has shown that considering a subject's prior fMRI (referred to as memory state) can enhance the accuracy of encoding current fMRI data. Consequently, we attempted to leverage prior fMRI representations (resting states), alongside image embeddings, as conditions to guide the synthesis of fMRI data, allowing MindSimulator to simulate the brain's activity transitioning from resting states to activated states. We have provided more explanations in the revised version to provide a better understanding.
>
>
> `Q2:`
>
> The primary reason is that **our encoding model cannot effectively generalize to fMRI data from other subjects**, due to the substantial difference between subjects’ brains. Obviously, the experiment subjects in the NSD dataset and the THING-fMRI dataset are **quite different**. Our encoding model trained on NSD subjects is **only capable of synthesizing fMRI for NSD subjects**. When applied to an out-of-distribution dataset, such as CIFAR-10 or CIFAR-100, the synthetic fMRI also corresponds to NSD subjects. Consequently, the image-fMRI pairs from other subjects, such as those from THING-fMRI subjects, cannot be used as ground truth for evaluation.
>
> **References**
>
> [1] Tal Yarkoni et al. Large-scale automated synthesis of human functional neuroimaging data. Nature methods, 2011, 8(8): 665-670.
>
> [2] Gia H. Ngo et al. Text2Brain: Synthesis of Brain Activation Maps from Free-Form Text Query. MICCAI 2021.
>
> [3] Huzheng Yang et al. Memory Encoding Model. arXiv:2308.01175v1.
>
> We sincerely appreciate your insightful feedback and thoughtful comments. We hope our response effectively addresses your concerns. Feel free to engage in further discussions, and your comments are greatly welcomed.
>
> Best wishes,
>
> All authors of Submission 180.

---

> > ### Comment · Reviewer_Atef · 2024-11-25
> > **Response to authors**
> >
> > Thank you for your detailed response, which helped me better understand your paper. However, I still have a few concerns:
> >
> > 1. **Regarding W1**: While I understand the rationale for using fLoc as a comparison method, this limits the analysis to a small number of concepts. You might consider comparisons with Grad-CAM-based methods, which allow for arbitrary concept selection. For example, see Shen G., et al., *Neuro-Vision to Language: Enhancing Brain Recording-based Visual Reconstruction and Language Interaction* (NeurIPS 2024).
> >
> > 2. **Regarding Q2**: My suggestion refers to training your encoding model from scratch on other datasets. This would help demonstrate that your method is not limited to NSD and is generalizable across other datasets.
> >
> > Overall, given these concerns, I will maintain my current score.

---

> ### Author Response · Authors · 2024-11-29
> **Replying to Reviewer Atef (Part-1)**
>
> Thanks very much for your feedback. We would like to further clarify your concerns.
>
> `W1:`
>
> Thanks for pointing out the Grad-CAM-based method for comparison. The Grad-CAM-based method is **unsuitable** as a baseline for voxel localization for three primary reasons.
>
> First of all, we briefly introduce the distinction between fLoc and Grad-CAM-based approaches concerning the localization of concept-selective voxels.
>
> - fLoc involves conducting **statistical analysis** (a voxel-wise one-sample t-test) on a **set of fMRI** corresponding to natural stimulus (e.g., images) of a particular concept to locate concept-selective voxels.
>
> - In contrast, Grad-CAM-based methods calculate the impact of voxels in **single fMRI input** on decoding a particular concept. This kind of approach generally involves text output or concept classification in fMRI decoding models to calculate concept-related gradients.
>
> Then, we provide the reasons as below:
>
> 1. Since our work performs concept localization based on the synthetic fMRI generated by our MindSimulator, **fLoc is introduced as an evaluation (rather than a comparative method) to verify the functional accuracy of the synthetic fMRI**. Specifically, we treat the concept localization results of fLoc as **ground truth** to evaluate the concept localization accuracy from our synthetic fMRI, thereby reflecting the effectiveness of the synthetic fMRI. Note that there is no recognized concept localization of brain regions, that is, there is actually no ground truth concept localization for evaluation. From this point, our method and Grad-CAM-based methods are incomparable. This is why the work you mentioned presents heatmap visualization and visual reconstruction results instead of quantitative results to validate the concept localization.
> 2. Additionally, the Grad-CAM-based method uses **single fMRI** for concept localization, which may be **coarse and questionable**. According to the paper you mentioned, it locates the concept-selective voxels in the early visual cortex, which contradicts the consensus in neuroscience. Typically, the early visual cortex is responsible for processing basic features such as color and shape, rather than specific concepts. Anyway, since there is no recognized concept localization of brain regions, we cannot say our concept localization is better than theirs. From the heatmap visualization, its concept localization results are quite different from ours, indicating the comparison between our method and Grad-CAM-based methods is meaningless and unfair.
> 3. Finally, the number of selective concepts of fLoc is limited by the given concepts in practical subject stimulation experiments, that is limited experimental data. Our work, similar to recent work on concept localization, e.g., Shen’s work, **aims to leverage generative models to extend concept localization to arbitrary concepts** (ideally), addressing the issue of limited experimental data. Differently, our work builds an image-to-fMRI generation model, aiming to simulate the human brain to generate quantitatively and functionally accurate fMRI. Then, we can utilize the generation models to synthesize fMRI corresponding to an arbitrary concept leveraging images of the concept. Grad-CAM-based methods generally adopt fMRI as input and learn to align fMRI to the semantic text or labels (i.e., concepts) of visual stimulus. Then, these methods capably visualize the “correlation” between fMRI and various concepts within the visual stimuli. Grad-CAM-based methods are also subject to the limited concepts in all visual stimuli. Accordingly, from the methodology of concept localization, our method is quite different from Grad-CAM-based methods, confirming it is unsuitable to compare ours with Grad-CAM-based methods.
>
> Given the aforementioned reasons, we did not compare our work with Grad-CAM-based methods. It is conceivable that in the future, these two approaches may converge towards a unified concept localization. However, at present, they remain distinctly incomparable.

---

> ### Author Response · Authors · 2024-11-29
> **Replying to Reviewer Atef (Part-2)**
>
> `Q2:`
>
> Thanks for your comments and suggestions. We regret not well introducing and explaining the out-of-distribution (OOD) generalization setting in our previous discussion, which may have led to your confusion and misunderstanding. Here, we introduce the motivation and setting of our OOD generalization experiments for your reference.
>
> **Motivation of the OOD experiments**:  As we discussed above, our work aims to simulate the human brain to generate quantitatively and functionally accurate fMRI for any visual stimulus and extend concept localization to arbitrary concepts. Accordingly, we introduce other image datasets (i.e., CIFAR used in the experiments) to verify the capability of our *MindSimulator* in synthesizing high-quality fMRI for the visual stimulus of OOD concepts. The OOD experiment is to explore the question: since the images used to train *MindSimulator* come from MSCOCO, can *MindSimulator* also accurately synthesize fMRI when images come from other datasets?
>
> **Setting of OOD experiments**: Based on the above motivation, we can outline the setting for OOD generalization experiments. First, we need an image-fMRI pair dataset $D_A$ to train the *MindSimulator*. Then, we synthesize fMRI using the images from dataset $D_B$ and evaluate the corresponding synthesis accuracy. This process is similar to zero-shot generalization. In our experiment, dataset $D_A$ is NSD, and $D_B$ is CIFAR-10/100. In the evaluation of OOD experiments, we verify the semantic-level accuracy of synthetic fMRI (Table 2) and visualize the reconstructed images from fMRI for validation. Additionally, note that our subject-specific *MindSimulator* is trained based on the image-fMRI pairs of a single subject. We did not intentionally select image-fMRI datasets for evaluation due to the fact that the substantial difference between subjects renders fMRI for other subjects are meaningless for comparison, which is the primary point we focused on in the last round of discussion.
>
> **Regarding the dataset generalization you mentioned**:
>
> - We first explain the reason for our settings. Since NSD is highly representative, many representative studies [1-6] have conducted experiments solely on the NSD dataset. Following previous studies, we also conduct experiments only on the NSD dataset.
>
> - THINGS-fMRI and NSD have a comparable number of image samples (8640 vs. 10000). Intuitively, our *MindSimulator* has the potential to effectively generalize to the THINGS-fMRI dataset.
>
> - Since we have not worked with this dataset before, it is difficult to complete the dataset generalization experiment within one week, especially considering the complicated fMRI preprocessing. However, we will implement it in the future and include it in the final version of our paper.
>
>
> **References**
>
> [1] Andrew F. Luo et al. Brain Diffusion for Visual Exploration: Cortical Discovery using Large Scale Generative Models. NeurIPS 2023.
>
> [2] Andrew F. Luo et al. Brain Mapping with Dense Features: Grounding Cortical Semantic Selectivity in Natural Images With Vision Transformers. arXiv:2410.05266.
>
> [3] Paul S. Scotti et al. Reconstructing the Mind's Eye: fMRI-to-Image with Contrastive Learning and Diffusion Priors. NeurIPS 2023.
>
> [4] Paul S. Scotti et al. MindEye2: Shared-Subject Models Enable fMRI-To-Image With 1 Hour of Data. ICML 2024.
>
> [5] Huzheng Yang et al. Memory Encoding Model. arXiv:2308.01175v1.
>
> [6] Hossein Adeli et al. Predicting brain activity using transformers. bioRxiv.

---

> > ### Comment · Reviewer_Atef · 2024-11-30
> >
> > Thank you for your detailed response. It has provided valuable insights and greatly improved my understanding of your work.
> >
> > 1. **W1**: I agree with much of your response regarding the limitations of Grad-CAM-based methods. Your independent fMRI generator is indeed a significant advance, as it does not rely on paired visual stimuli and fMRI for concept localization. This independence makes your method more flexible and scalable. You also noted that some results in Shen et al.'s paper may conflict with neuroscientific consensus, which is a valid concern. However, a direct comparison of concept localization results between your method and Shen et al.'s work on the same concepts could further strengthen your claims. Such a comparison could demonstrate better alignment with established neuroscientific knowledge and provide a valuable baseline for evaluation.
> >
> > 2. **Q2**: Thank you for clarifying the motivation and setup of your OOD experiments. Generalization across diverse datasets is critical for demonstrating the robustness of your algorithm. While I understand the challenges of implementing a generalization experiment within a short timeframe, I appreciate your commitment to including such experiments in the final version of your paper.
> >
> > In summary, thank you again for your effort in addressing my concerns, and I will increase my rating.

---

> > > ### Author Response · Authors · 2024-12-01
> > > **Respond to Reviewer Atef**
> > >
> > > We sincerely thank you for taking the time to patiently read through our feedback. We are even more grateful for the constructive advice you have provided and for the recognition and positive evaluation of our manuscript. Following your suggestions, we will further improve our manuscript:
> > >
> > > + We will use our method to locate the concept-selective regions mentioned in Shen et al.'s paper and conduct a direct qualitative visual comparison, along with supplementary analysis. As you mentioned, such a comparison not only demonstrates that our method aligns more closely with established neuroscience knowledge, but also provides a valuable baseline for evaluation.
> > > + We will train our MindSimulator using the THINGS-fMRI dataset to further verify the generalization of our method across various datasets.
> > >
> > > We promise that we will update the above improvements in the final version. Thank you again for your time and effort!

---

### Official Review · Reviewer_SRDC · 2024-11-01

**Soundness:** 3
**Presentation:** 3
**Contribution:** 3
**Rating:** 6
**Confidence:** 4

**Summary:**

This paper proposes a new way to implement concept-localization in the brain using a learned generative model which synthesizes fMRI responses. This is derived from the observation that fMRI responses to the same stimuli can be noisy and are better captured by sampling from a random variable instead of a learned (discriminative/static) model. A latent representation is jointly learned via CLIP in which an image embedding is paired with a voxel embedding and then trained according to the SoftCLIP loss.

The authors show reconstruction is possible via their proposed method to use synthetic fMRI, but the authors fail to show that the brain data could just as easily be ignored. Image encodings are passed into the sampler and decoders are highly able to create very realistic images but it's not clear that the modelling of the resting state inputs and learned fMRI is actually doing anything useful in a clear way as the authors make it seem (with huge swaths of cortex claimed for very restrictive conceptual categories). There are arguments for how discrete these are but once you expand the classes beyond the very limited amount presented, then quantified overlap, it would be clear that many patches are not conceptually distinct. That's my assumption.

The idea is an interesting one, but the lack of good experimental testing against strong baselines (particularly, testing with shuffled/random fMRI data). The bulk of the promise shown here might actually be just by going between image embeddings (via the voxel encoder as it was jointly trained on image representations and not fMRI data alone).

**Strengths:**

The paper has a pretty good grasp on the recent literature and various approaches that have been experimented on in this area, showing a wide depth of knowledge. The analyses seem detailed and it's clear a lot of work went into some parts of the experimental analysis. There will be a pretty detailed Weaknesses section, but it's easier to point out identified weaknesses than identify lists of things done correctly. I do have a fair few issues with the way the analysis was done, but I think with some tweaks and additional analyses that are robust against better controls, better description, this paper does have potential.

**Weaknesses:**

* Captions to figures are mostly vacuous and non-descriptive and need to be expanded to better describe the associated figures
* Some points are argued but are presented without evidence and for the kind of statements they are, definitely require a solid backing (see Questions section)
- Citation format is not consistent. Many citations should be in parentheticals but are not (needs to be fixed for camera-ready version)
- The language is overly flowery in a way that makes the claims nonsensical (e.g. “Fortunately, we effectively explore novel concept-selective regions, capably providing explicit hypothesis constraints…”)
- Language needs to be checked by a person intimately familiar with the conventions of academic written English to correct some unusual and unclear phrasing (in the methods section especially)

There have been numerous recent works that have highlighted how these types of models can effectively perform the same function when replacing brain data with random noise or brain responses that aren't paired correctly with the same responses.

Huge caveat here that “**if it can be reconstructed, then the fMRI contains the information**” but you can often do reconstruction equally well from random noise, there doesn’t have to be anything real in the fMRI data. Kamitani recently showed this (https://arxiv.org/abs/2405.10078) and this paper also did (https://arxiv.org/abs/2405.06459) with EEG.

The paper fails to take into account a number of confounds and does not seem to understand just how drastic this aspect of the analysis could be on changing the presented results. You can't present images of food and not take into account that you might be modelling shape (round plates, round food shapes) or lower-level features like colour (food is often colourful). These have been huge issues in the concept localization space using datasets like NSD but I didn't see any citations or awareness of this issue. Also, it's not likely that the concept of "surfer" or "bed" takes up anywhere near as much cortical territory as some of these plots indicate. There is high-level confounding going on here that undermines the idea of concept localization. This is why the handcrafted stimuli were carefully created in the first place, to avoid this issue. The idea of this paper seems like it goes back in the wrong direction.

The results in Table 3 during the ablation analysis show often minimal drops when ablating important components of the paradigm, which lead me to believe that confounds and lack of good baselines are hiding shortcut learning and cheats that the model is making use of instead of it being primarily a method centred on good fMRI representations.

If you have focused on the localizers used in NSD then I think it's important you cite the (ubiquitous) paper that NSD (and many other fMRI datasets) use, namely the fact that these fLoc images come from Stigliani et al. 2015 (https://www.jneurosci.org/content/35/36/12412).
It seems quite the oversight to not have cited this given the content of the submission, especially as you're using the images from this paper in the figures of your dataset.

**Questions:**

The introduction raises some claims that need supporting evidence. How do you know that the efforts to go into designing common functional localization images are insufficient and poorly generalizable? What promotes this observation? Why should a functional localiser be embedded within a naturalistic scene? Naturalistic stimuli are famously confounded across multiple dimensions and the artificial placement of a core concept in a bare background is a method to remove potential confounds. Yes, it’s undesirable because our vision is based around naturalistic scenes, but the argument for naturalistic images in functional localization is only inviting trouble. We would lose specificity and be more unlikely to be sure that we’re not detecting confounding background information and mistakenly attributing brain activity to core concepts within the images. The arguments as they’re outlined don’t naturally follow on from one another in this exposition of the paper’s contributions.

- Line 157: do you mean for the comma to be a subtraction symbol in the MSE equation?
- Why do you start the inference sampler with resting state fMRI data? What's the idea here? It's not really explained in Section 3.4.
- If an amended version is submitted, could you put the ROI boundaries on your flatmaps to better orient the distributions of voxel encodings?
- In 6.1 what's going on here? Are you using MSCOCO images or images for which there is fMRI data in NSD? It's not clear
- Also in 6.1, you mention the t-test that is done voxelwise, where are these results? What was the threshold? I don't really understand what you've done or how you have set up your test and there is also no mention of multiple comparisons correction (big red flag for me).

**Details Of Ethics Concerns:**

N/A. Public datasets used.

---

> ### Author Response · Authors · 2024-11-24
> **Author Rebuttal for Reviewer SRDC (Part-1)**
>
> Dear Reviewer SRDC,
>
> Thanks for your constructive comments and valuable advice. We will address your concerns one by one.
>
> `W1:`
>
> Thank you for your suggestions, we have included appropriate captions for **each table and figure** to enhance the clarity and comprehensiveness of their descriptions.
>
> `W2:`
>
> We have carefully revised the statement of the claims you mentioned. Please refer to our response `Regarding our claims in Introduction` below.
>
> `W3:`
>
> Thank you for pointing out the citation formatting issue. We have carefully **examined all citations** and made corresponding revisions in the revised version.
>
> `W4:`
>
> **We have modified the presentation of the "Introduction" section** to make it more rigorous. If there are any other shortcomings, please do not hesitate to point them out.
>
> `W5:`
>
> Thank you for your suggestion. We have made a careful proofreading to improve the quality of the paper writing.
>
> `Regarding the random noise as input:`
>
> I have carefully read the two papers you provided.
>
> + Prof. Kamitani's paper does not mention that visual semantics can be decoded from pure noise. **He argues that current research on neural decoding has overstated its reconstruction performance**, and the generalization, authenticity, and stability of decoding models need further improvement.
> + The second paper does report that linguistic information can be decoded from noise. We speculate this is because **its EEG-to-text decoding model essentially fine-tunes pre-trained large language models using small datasets**. This fine-tuning process can somehow overfit limited noise data. In contrast, **similar issues might not exist in fMRI visual decoding models** like MindEye2, since fMRI-to-image embedding generation is trained from scratch.
>
> Based on the literature you provided, we acknowledge that our statement, "If it can be reconstructed, then the fMRI contains the information," lacks the necessary rigor. **Indeed, we should provide some premises or constraints**. To clarify, successful decoding can only suggest that the fMRI holds the relevant visual information when the testing data aligns with the generalization capacity of the trained visual decoding model. **We have refined the statement in the revised manuscript**.
>
> To further address your concerns, **we conducted experiments replacing fMRI with pure noise as the input to the trained fMRI visual decoding model**. The results, detailed in our **Appendix B.3**, demonstrate that **no clear visual semantics can be reconstructed from noise**.
>
> `Regarding the natural stimuli:`
>
> Thank you for sharing your insightful perspective. Your concerns are valid and thought-provoking. We would like to gently point out that you may **overlook the benefits brought by the large-scale fMRI available**. Indeed, when dealing with a limited amount of stimulus and corresponding fMRI data, the intricacies of natural scenes can introduce irrelevant factors that might impede the accuracy and efficacy of concept localization. In such scenarios, manually excluding these irrelevant factors could prove advantageous. However, **when utilizing the vast fMRI data generated by MindSimulator, the potential exists to mitigate the influence of the irrelevant factors**. Taking food images as an example, due to the large amount of food images, the image collection covers a variety of colors. Also, the probability of repetitive activations of voxels in the lower-level visual cortex associated with specific colors is low, while all activations are associated with specific food concept. Subsequent to the t-test and threshold filtering, **only the consistently activated food-selective voxels are selected and localized**, instead of the color-related voxels. Therefore, leveraging large-scale synthetic fMRI could effectively minimize the impact of irrelevant factors in natural scenes. Based on these considerations, we argue that visual stimuli of natural scenes can indeed be utilized to explore the localization of concepts. **Certainly, the above considerations require further solid evidence and exploration, which is an emerging research field attracting increasing attention and precisely our future research direction**.

---

> ### Author Response · Authors · 2024-11-24
> **Author Rebuttal for Reviewer SRDC (Part-2)**
>
> `Regarding the performance:`
>
> + **Quantitative results**: Note that, after ablating key components, the corresponding performance shows **a significant decrease** in terms of voxel-level metrics such as MSE and Pearson. However, semantic-level metrics have relatively high values but low differences **owing to their calculation method**. For instance, the four metrics—Alex(2), Alex(5), Incep, and CLIP—are based on binary classification accuracy, with a minimum value of 50%. That is, as long as the images contain certain semantic information, the classification accuracy will reach a relatively high level.
>
> + **Baselines**: Due to the limited research focusing on fMRI encoding, there are **fewer suitable baseline methods for comparison**. Our **linear model baseline** essentially represents encoding models **widely used in neuroscience research**. The **Transformer encoding model** we adopted follows the structure and training process in reference [3]. Notably, the encoding model from [3] achieves desirable performance (**2nd** place) in the **Algonauts Challenge [4]**, signifying its **superiority to the majority of methods** employed in the fMRI encoding challenge. As a result, **the baselines selected for the experiments are representative**, yielding convincing experimental results.
>
> + **Better fMRI representation**: **The goal of our paper is to enhance the synthesis quality of fMRI**. To achieve this, we have made every effort to find suitable baseline methods and comprehensive metrics for evaluation. We have also explored diverse approaches to enhance the performance of our generative encoding model. Accordingly, we argue our extensive experiments verify the effectiveness of our encoding model.
>
> `Regarding the missing references:`
>
> Thank you for pointing out the key reference. Neglecting to cite the original fLoc paper is indeed an oversight on our part. **We have cited the reference in the revised version**.
>
> `Regarding the claims in our Introduction:`
>
> Our wording in the introduction section may lead to the **misunderstanding**. **Utilizing the fLoc experiment for conceptual localization is accurate and effective**. What we refer to as "**insufficient**" pertains to **the number of concepts**. Since fLoc relies on specially curated visual stimuli that require manual processing, such datasets are rare. Therefore, **when attempting to localize a less-studied concept, it is highly likely that there will be few or no directly available images**. Similarly, our reference to "**generalizability**" stems from this issue: without directly available data, **fLoc is difficult to generalize to the localization of new concepts**. In contrast, natural scenes offer rich visual stimuli in terms of both quantity and semantics, potentially providing better generalizability. Regarding the effectiveness of natural stimuli, we have discussed in the above.
>
> In addition, our proposed strategy of using MindSimulator to synthesize fMRI data for concept-selective localization is not intended to replace fLoc. Instead, **we regard it as a complementary and auxiliary approach to fLoc**, particularly in scenarios where manually constructing numerous visual stimuli and collecting corresponding fMRI data pose challenges.

---

> ### Author Response · Authors · 2024-11-24
> **Author Rebuttal for Reviewer SRDC (Part-3)**
>
> `Q1:`
>
> The misuse of the comma is our oversight, and we have corrected the error in the revised version.
>
> `Q2:`
>
> We used noised resting-state fMRI representations as initialization for iterative denoising. **This design was introduced to allow MindSimulator to simulate the brain's activity transitioning from resting states to activated states**. Recent research [1] in fMRI encoding has shown that considering a subject's prior fMRI (referred to as memory state) can enhance the accuracy of encoding current fMRI data. Consequently, we attempted to leverage prior fMRI representations (resting states), alongside image embeddings, as conditions to guide the synthesis of fMRI data.
>
> Note that subjects viewed different images consecutively in the NSD experiment. We have also experimented with utilizing noised prior fMRI representations instead of pure noise as a starting point of the iterative denoising process. Regrettably, these approaches proved ineffective, which may be caused by the complexity of fMRI data collection experiments or the intricate mechanisms involved in the transition of brain activity. Finally, we chose to use noised resting-state fMRI as the inference initialization. We have provided more explanations in the revised version to provide a better understanding. Although it lacks further exploration and sufficient explanations of the effect of resting-state brain activity fMRI, **it is intriguing and worthwhile to explore the transition of brain activity in brain encoding/decoding analysis**.
>
> `Q3:`
>
> Certainly, we have put the ROI boundaries on our flat maps. Please refer to the **Figure 8** of our revised version. Thanks for your valuable suggestion.
>
> `Q4:`
>
> In Section 6.1, **we used the MSCOCO images in the NSD experiment** (a total of 73,000 images, including 9,000 images unique to each subject and 1,000 images shared among subjects). Note that each subject had fMRI data corresponding to only 10,000 out of 73,000 images. We have clarified this in **Appendix A.4** of our revised version.
>
> `Q5:`
>
> In **Appendix A.4**, we provide the **t-test threshold**, set to 2 following the settings in [2]. We will further provide the **t-test results** in our code repository. Moreover, **the results reported in our paper are averaged values across 3 comparisons** under different random seeds and we will clarify it in **Table 4** of the revised version.
>
> **References**
>
> [1] Huzheng Yang et al. Memory Encoding Model. arXiv:2308.01175v1.
>
> [2] Andrew F. Luo et al. Brain Diffusion for Visual Exploration: Cortical Discovery using Large Scale Generative Models. NeurIPS 2023.
>
> [3] Hossein Adeli et al. Predicting brain activity using transformers. bioRxiv.
>
> [4] Alessandro T. Gifford et al. The algonauts project 2023 challenge: How the human brain makes sense of natural scenes. arXiv:2301.03198.
>
> We sincerely appreciate your insightful feedback and thoughtful comments. We hope that our response effectively addresses your concerns. Feel free to engage in further discussions, and your comments are greatly appreciated.
>
>
> Best wishes,
>
> All authors of Submission 180.

---

> > ### Comment · Reviewer_SRDC · 2024-11-26
> > **Thank you for the response**
> >
> > I am quite impressed with the effort the authors have put into responding the reviewer (+ public) comments on this paper. It's clear a lot of effort went into this response and to some extent some of my concerns have been assuaged. However, some main concerns of mine remain unaddressed. These primarily relate to the lack of multiple comparisons correction in the voxel-wise statistical tests. Nothing seems to have been given as a response to this question. Can the authors point to the reference to support that Stigliani et al.'s threshold value is zero? This seems odd to me.
> >
> > I was also most skeptical about the use of resting state values as this does not seem justified to me. The supporting reference to the Algonauts winner is indeed interesting and we all saw that when the paper was posted, but this was memory encoding from delayed haemodynamic response activity. It was lingering information from the viewing of prior images in the experiment. This isn't connected to what might be beneficial in matching with resting state data. I see the link you've made, but successful use of the prior information is literally from trials in the immediate seconds paired with image information, not some abstract fMRI signal potentially from a different session.
> >
> > There is no a priori reason to expect that this better conditions the model to generate better images as there is no information from the stimuli present. In Yang et al. there was a clear mechanistic understanding that information from prior trials in the preceding seconds made for better voxel encoding procedures. This link is a bit too tenuous here to make it strongly motivated.
> >
> > I should have mentioned this earlier, but it only just struck me after discussions with other researchers over the past few weeks. There really should be better baselines, for example in Table 1, I'd want to see the voxel-wise correlation between shuffled responses to determine what is potentially explained by looking at low-signal generic fMRI responses to images in general. I think you'd detect a pretty close value there and it would potentially raise some questions about the nature of the signal you are capturing.
> >
> > I really am thankful for the efforts made to address reviewer concerns, but I am still quite uncomfortable with some of these results and I don't quite think it's ready for the seal of approval that acceptance here would give this work. That's why I will keep the same rating. I do look forward to seeing this work again with better quality checking and better baselines, better analysis of the contribution of what's in the resting state data to drive potentially beneficial effects.

---

> > > ### Author Response · Authors · 2024-11-30
> > > **Replying to Reviewer SRDC (Part-1)**
> > >
> > > Thanks for your detailed feedback and insightful discussion. We address your remaining concerns point by point as follows.
> > >
> > > `Regarding multiple comparisons correction:`
> > >
> > > First, we would like to express our regret that we did not focus on multiple comparisons correction since we are not familiar with multiple comparisons correction and failed to fully understand your suggestion.
> > >
> > > Indeed, in our previous concept-selective region localization, we did not use the multiple comparison correction. Following your suggestion, we conducted the one-sample t-test and applied the **Bonferroni correction** (the most commonly used approach for multiple comparison adjustment) with a significance level of 0.01 to localize the concept-selective voxels. Subsequently, we validated the concept-selective voxels identified by the linear encoding model and our MindSimulator against the Ground Truth obtained from the NSD fLoc experiment. The according results of localization accuracy and F1 scores are provided as follows:
> > >
> > > | Places-selective | ACC |     | F1  |     | Bodies-selective | ACC |     | F1  |     |
> > > | --- | --- | --- | --- | --- | --- | --- | --- | --- | --- |
> > > | # Images | Linear | Ours | Linear | Ours | # Images | Linear | Ours | Linear | Ours |
> > > | Top 100 | 32.0% | **52.6%** | 0.461 | **0.573** | Top 100 | 45.8% | **91.7%** | 0.570 | **0.674** |
> > > | Top 200 | 30.0% | **45.6%** | 0.444 | **0.563** | Top 200 | 42.1% | **84.2%** | 0.551 | **0.727** |
> > > | Top 300 | 29.2% | **41.3%** | 0.437 | **0.540** | Top 300 | 40.7% | **81.4%** | 0.543 | **0.740** |
> > > | Top 500 | 28.5% | **37.3%** | 0.430 | **0.513** | Top 500 | 39.2% | **75.3%** | 0.532 | **0.732** |
> > > | Top 1000 | 27.9% | **33.5%** | 0.425 | **0.483** | Top 1000 | 37.9% | **64.2%** | 0.523 | **0.693** |
> > >
> > > The evaluations in the above Table are slightly different from those of our previous uncorrected experiments. The results still verify our claim that the fMRI synthesized by our MindSimulator is more accurate than the fMRI synthesized by linear models in terms of concept localization.
> > >
> > > Kindly remind that we failed to update the results in the current revised version since the deadline of updating revised version was past. To guarantee convincing experiments, we promise to update the results with the new results in the final version.
> > >
> > > `Regarding the NSD fLoc threshold:`
> > >
> > > NSD used the stimulus set provided by Stigliani et al. and set the t-value threshold to 0 in determining face-, word-, body-, and place-selective regions. The settings can be referred to the NSD official website ( https://cvnlab.slite.page/p/X_7BBMgghj/ROIs ), which mentions that “*These ROIs were the result of (liberal) thresholding at t > 0*”.
> > >
> > > `Regarding resting-state fMRI:`
> > >
> > > Thanks for your constructive discussion. The previous work reflects the memory encoding effect from delayed hemodynamic response activity, which is not sufficient to support the utilization of resting state fMRI. The introduction of our resting-state fMRI to enhance voxel encoding is indeed not well-justified. However, we would like to kindly remind you that the introduction of resting-state fMRI is not the main contribution of the key aspect of our work.
> > >
> > > Additionally, as we mentioned in the last round of discussion, we have conducted experiments to compare different initializations with pure noise, noised resting-state fMRI, and noised prior fMRI. The results are provided in the following table, which is also shown in Appendix B.6. From the results, we can observe that both the resting-state fMRI initialization and prior fMRI initialization do not show priority over pure noise initialization. In view of the experimental results, we will tentatively remove the design of resting-state fMRI initialization in the final version, to avoid confusing design. In future works, we will try our best to explore the transition of brain activity in brain encoding/decoding analysis.
> > >
> > > | Methods | Pearson↑ | MSE↓ | PixCorr↑ | SSIM↑ | Alex(2)↑ | Alex(5)↑ | Incep↑ | CLIP↑ | Eff↓ | SwAV↓ |
> > > | --- | --- | --- | --- | --- | --- | --- | --- | --- | --- | --- |
> > > | Pure Noise | 0.322 | 0.418 | 0.204 | 0.302 | **90.9%** | 96.5% | **93.0%** | 89.8% | 0.712 | 0.407 |
> > > | Noised Resting-state fMRI | **0.326** | **0.417** | 0.207 | **0.305** | 90.6% | **97.1%** | 92.8% | 89.8% | **0.714** | 0.402 |
> > > | Noised Previous fMRI | 0.325 | 0.420 | **0.209** | 0.302 | 90.3% | 97.0% | 92.3% | **90.0%** | **0.714** | **0.400** |

---

> > > ### Author Response · Authors · 2024-11-30
> > > **Replying to Reviewer SRDC (Part-2)**
> > >
> > > `Regarding shuffled responses as baselines:`
> > >
> > > Based on your suggestions, we have added the corresponding experiments as the baselines of Table 1, including the following:
> > >
> > > - **Random Input**: Each voxel is independently sampled from a standard Gaussian distribution.
> > >
> > > - **Shuffled GT fMRI**: The voxels of each ground truth fMRI are randomly shuffled.
> > >
> > > - **Shuffled Synthetic fMRI**: The voxels of each fMRI synthesized by MindSimulator are randomly shuffled.
> > >
> > >
> > > After shuffling, the fMRI is input into the trained decoding model (MindEye2), and the reconstruction results are obtained. Further voxel-level and semantic-level evaluation metrics are calculated as follows:
> > >
> > > | Methods | Pearson↑ | MSE↓ | PixCorr↑ | SSIM↑ | Alex(2)↑ | Alex(5)↑ | Incep↑ | CLIP↑ | Eff↓ | SwAV↓ |
> > > | --- | --- | --- | --- | --- | --- | --- | --- | --- | --- | --- |
> > > | Random Input | 0.0002 | 1.430 | 0.027 | 0.178 | 50.7% | 50.0% | 50.8% | 50.1% | 0.980 | 0.632 |
> > > | Shuffled GT fMRI | 0.0004 | 0.802 | 0.023 | 0.143 | 50.4% | 51.0% | 50.4% | 50.2% | 0.983 | 0.642 |
> > > | Shuffled Synthetic fMRI | 0.0011 | 0.612 | 0.021 | 0.114 | 50.4% | 51.4% | 49.9% | 50.7% | 0.986 | 0.650 |
> > > | MindSimulator (Ours) | **0.3570** | **0.385** | **0.202** | **0.298** | **89.7%** | **97.0%** | **93.1%** | **91.2%** | **0.689** | **0.391** |
> > >
> > > It can be seen, after the fMRI is shuffled, its decoding performance is comparable to that of random input. **Most performance of baselines reach their theoretical minimum values**, such as Pearson correlation and pixel similarity (PixCorr) approaching 0; Alex(2), Alex(5), Incep, and CLIP nearing 50%.
> > >
> > > These results suggest that **trained decoding models can successfully capture the visual semantics contained in fMRI rather than other irrelevant content**. At the same time, this also indicates that **reconstruction can only be achieved when the fMRI contains the real visual semantics**. I think these results should further alleviate your concerns about the trained fMRI visual decoding model.

---

> > > > ### Comment · Reviewer_SRDC · 2024-11-30
> > > > **Response**
> > > >
> > > > I thank the authors for their extra work they have put into the response. I do still have some reservations but I really appreciate the understanding and admission of the authors that the prior aspects of linking to previous studies were not directly in full support of the conclusions (the admission that resting state fMRI is not well-justified) but as the authors say, this is indeed not the primary contribution of the paper. I would be interested in exploring the code and trying to replicate some of these results myself.
> > > >
> > > > I have warmed in my opinion of this submission since first reading it, mainly due to the comprehensive nature and productive discussion in the rebuttal with the reviewers. I have updated my Soundness score to 3, my Contribution score to 3 and I have raised my rating to 6 in recognition of the positive efforts of the authors to tackle the reviewer doubts.

---

> > > > > ### Author Response · Authors · 2024-12-01
> > > > > **Respond to Reviewer SRDC**
> > > > >
> > > > > We sincerely appreciate your continuous involvement in the discussion of our paper and the highly professional suggestions you provided. I am especially grateful for the recognition and positive feedback you gave on our manuscript. After making revisions based on your advice, our paper has seen significant improvements in terms of expression accuracy, methodological soundness, and experimental rigor. Thank you again for your time and effort!

---

### Official Review · Reviewer_KVAs · 2024-11-03

**Soundness:** 1
**Presentation:** 1
**Contribution:** 1
**Rating:** 5
**Confidence:** 2

**Summary:**

This is a paper with interesting results. However, I am uncertain whether the technique presented represents a major advancement. A key gap in the literature appears to be the investigation of cognitive processes, including the concepts discussed by the authors, not only in a spatial context but also in terms of activation trajectories over time. The authors have already addressed this point in their discussion with another reviewer. While I agree that NSD does not necessarily contribute additional insights in this context, there are datasets on movie viewing with corresponding ratings that could have been beneficial for this study.

**Strengths:**

-

**Weaknesses:**

A key gap in the literature appears to be the investigation of cognitive processes, including the concepts discussed by the authors, not only in a spatial context but also in terms of activation trajectories over time. I am not sure if the current paper provides a systematic solution.

**Questions:**

Nothing anymore. the authors have done a great job.

**Details Of Ethics Concerns:**

All fine

---

> ### Public Comment · ~Andrew_Luo1 · 2024-11-13
>
> I am one of the authors of BrainDiVE.
>
> I wanted to quickly provide my thoughts for `MindSimulator: Exploring Brain Concept Localization via Synthetic fMRI`.
>
> I have carefully read this paper and I will summarize it as follows:
> 1. The authors train an fMRI autoencoder with a latent space that is regularized by CLIP
> 2. They train an diffusion transformer which synthesizes fMRI responses conditioned on the CLIP image embedding
> 3. They take the expectation of the fMRI response by sampling from the diffusion model multiple times, and then taking an average.
>
> Pros:
> 1. I think the approach overall of a diffusion fMRI encoder is quite novel. Indeed the fMRI response is stochastic, so this design does make sense to me.
> 2. Without the use of gradients, computational tests of selectivity can be done much more quickly.
>
> Questions & weaknesses:
> 1. I'm a bit unsure about the proposed evaluation metrics for fMRI encoding performance. Using pearson R or R^2 is standard in fMRI encoder literature. Using a decoder as part of the evaluation process introduces additional complications.
> 2. Using `Resting-State Brain Activity fMRI` as the inference initialization is a bit strange, in my view this is not well justified
> 3. Using `Correlated Gaussian Noise` as the multi-trial seed is not well justified.
> 4. The fMRI beta pre-processing stage is a bit unclear. Do the authors use all three repeats of the same image individually? Or do the authors average the beta values?
>
>
> Minor issues:
> 1. Typo in Figure 4 left text? `Trial = 1` seems to be repeated twice
>
> If I were the reviewer of this paper, I would give this paper a 6 (weak accept) prior to rebuttal.

---

> > ### Author Response · Authors · 2024-11-16
> > **Author's response to public comments**
> >
> > Dear Prof. Luo,
> >
> > Thank you for your valuable feedback as the primary author of BrainDIVE. Your direct comments are essential in distinguishing BrainDIVE from our MindSimulator. Your acknowledgment of MindSimulator's novelty and contribution has truly motivated us. We are currently addressing all reviewers' comments and yours, and plan to submit our responses and the revised manuscript as soon as possible.
> >
> > Thank you again for your comments.
> >
> > Best wishes,
> >
> > All authors

---

> > ### Author Response · Authors · 2024-11-24
> > **Author Rebuttal for Prof. Andrew Luo**
> >
> > Dear Prof. Luo,
> >
> > Thank you for your constructive comments. We will address your concerns point by point.
> >
> > `Q1:`
> >
> > While using decoding models to evaluate the quality of fMRI encoding introduces additional complications, **this computational burden is acceptable**. Experimental results on a single NVIDIA Tesla V100 GPU show that the average time cost for evaluating a single fMRI is within 1-2 seconds. We have included the time costs of the diffusion fMRI decoder for clarity in **Appendix A.2** of the revised version.
> >
> > In addition, our evaluation approach offers **two notable advantages**.
> >
> > + On one hand, this approach can **evaluate the semantic accuracy** of synthetic fMRI data. Previous metrics have primarily focused on localized voxel-level encoding performance, which is inadequate as visual semantics are predominantly embedded in global response patterns (as elaborated in **lines 257-269**).
> >
> > + On the other hand, employing the decoding model **facilitates a more intuitive evaluation**. Through decoding fMRI, we are able to conduct fMRI visualizations and evaluate the quality of synthetic fMRI via direct visual inspection—a functionality that is absent in conventional evaluation metrics.
> >
> > In addition, **leveraging established trained models to evaluate the semantic accuracy of generated content is a well-recognized strategy**. For instance, the FID metric, commonly used in the evaluation of image generation, employs Google's Inception V3 [1] for semantic evaluation. Similarly, in neuroscience, the acclaimed MindEye [2,3] shows potential as a semantic evaluation tool for synthetic fMRI data.
> >
> > `Q2:`
> >
> > The approach of utilizing resting-state brain activity fMRI as the inference initialization is introduced as **an exploration to incorporate the temporal dependencies of fMRI**. Recent research [4] in fMRI encoding has shown that considering a subject's prior fMRI (referred to as memory state) can enhance the accuracy of encoding current fMRI data. Consequently, we attempted to leverage prior fMRI representations (resting states), alongside image embeddings, as conditions to guide the synthesis of fMRI data, **allowing MindSimulator to simulate the brain's activity transitioning from resting states to activated states**.
> >
> > Note that subjects viewed different images consecutively in the NSD experiment. We have also experimented with utilizing noised prior fMRI representations instead of pure noise as a starting point of the iterative denoising process. Regrettably, these approaches proved ineffective, which may be caused by the complexity of fMRI data collection experiments or the intricate mechanisms involved in the transition of brain activity. Finally, we chose to use noised resting-state fMRI as the inference initialization. We have provided more explanations in the revised version to provide a better understanding. **Although it lacks further exploration and sufficient explanations of the effect of resting-state brain activity fMRI, it is intriguing and worthwhile to explore the transition of brain activity in brain encoding/decoding analysis**.
> >
> > `Q3:`
> >
> > The idea of using Correlated Gaussian Noise as multi-trial seeds is motivated by recent work on video generation [5]. This research highlighted that using Correlated Gaussian Noise as multi-frame seeds for frame reconstruction significantly heightened the resemblance between video frames, preserving nuanced distinctions and leading to more seamless video reconstruction. Our goal is also to **generate multi-trial synthetic fMRIs that are highly similar to each other since averaging similar fMRI can better enhance voxel-wise reproducibility**. Therefore, we adopt Correlated Gaussian Noise.
> >
> > `Q4:`
> >
> > During the **training** phase, we used **all three repeats** of the same image individually. During the **testing** phase, we **averaged the ground-truth beta values** as the target. We will clarify the setup of the dataset in **Section 4.1** in the revised version.
> >
> > `Minor:`
> >
> > Thanks for pointing out the typo error. In **Figure 4**, the last row should be "**Trial = 5**". We will correct it in the revised version.
> >
> > **Reference**
> >
> > [1] Christian Szegedy et al. Rethinking the Inception Architecture for Computer Vision. CVPR 2016.
> >
> > [2] Paul S. Scotti et al. Reconstructing the Mind's Eye: fMRI-to-Image with Contrastive Learning and Diffusion Priors. NeurIPS 2023.
> >
> > [3] Paul S. Scotti et al. MindEye2: Shared-Subject Models Enable fMRI-To-Image With 1 Hour of Data. ICML 2024.
> >
> > [4] Huzheng Yang et al. Memory Encoding Model. arXiv:2308.01175v1.
> >
> > [5] Songwei Ge et al. Preserve Your Own Correlation: A Noise Prior for Video Diffusion Models. ICCV 2023.
> >
> > We sincerely value your insightful feedback and thoughtful discussion. We hope that our response effectively addresses your concerns. Feel free to engage in further discussions, and your comments are greatly appreciated.
> >
> > Best wishes,
> >
> > All authors of Submission 180.

---

> > > ### Public Comment · ~Andrew_Luo1 · 2024-11-25
> > > **Response to "Author Rebuttal for Prof. Andrew Luo"**
> > >
> > > I thank the authors for the detailed response.
> > >
> > > I have the following questions that I would like clarifications on:
> > > 1. `Q1` would it be possible for the authors to use a more traditional color map for $R^2$ visualization on page 17? (Such as matplotlib `hot`). I just want to see if the method is largely comparable with traditional methods (linear) in terms of accuracy.
> > > 2. I suggest the authors eventually add an ablation regarding the resting state experiment. Could the authors also clarify the timestep ratio for resting state initialization? I think it is possible resting state initialization could help, but I believe it would not necessarily help significantly.
> > > 3. I am not convinced by the justification in Q3. For video generation -- video has inherent temporal correlation structure. However for beta values you are explicitly removing the temporal axis by GLM regression. I think if the method could estimate a correct conditional distribution p(brain | image) with a gaussian error, correlated noise would not be needed. So in my view -> either your method can really estimate the correct conditional, and you don't need correlated noise; Or there is something likely wrong in the diffusion estimation?
> > >
> > >
> > > Stronger evidence for the design (resting state initialization) + details on the timestep used for resting state initialization would be helpful. I'm also not fully convinced by the correlated noise.
> > >
> > > As I am not a reviewer, so I cannot comment on whether or not this paper should be accepted or not.
> > >
> > > Overall I think this is an interesting paper and an **interesting approach**, and I think the authors have indeed proposed something novel and useful.

---

> ### Author Response · Authors · 2024-11-24
> **Author Rebuttal for Reviewer KVAs**
>
> Dear Reviewer KVAs,
>
> There are distinct differences between our paper and BrainDIVE. We outline the key disparities in terms of methodology, evaluation, and exploration of neuroscience for your reference.
>
> `Regarding the methodology:`
>
> Typically, to generate a synthetic fMRI, three key components are required: an input **image**, an image **embedder** like ViT, and an **encoding model** that projects the image embedding to fMRI data.
> + **BrainDIVE concentrates on refining the input images**, enhancing their quality by fine-tuning Stable Diffusion conditioned on specific brain regions with maximum activation.
> + In contrast, **our focus lies on optimizing the encoding model**. We introduce a generative encoding model instead of BrainDIVE's Linear model, aiming to enhance accurate encoding. Intuitively, our model can collaborate with BrainDIVE to enhance the overall quality of fMRI encoding.
>
>
> `Regarding the evaluation:`
>
> Overall, the evaluations of our work and BrainDIVE are significantly different.
> + BrainDIVE **evaluates images** to check if the generated images can notably activate specific brain regions.
> + In contrast, our focus lies on **evaluating fMRIs** to compare the accuracy of synthetic fMRIs generated by our MindSimulator against baselines at both voxel and semantic levels.
>
> `Regarding the neuroscience exploration:`
>
> While our work and BrainDIVE share the goal of locating concept-selective regions in the brain, our strategies differ fundamentally.
> + BrainDIVE localizes concept-selective regions by generating images that activate specific brain areas, followed by analyzing image semantics to deduce the related concepts.
> + In contrast, our method initiates by selecting images according to predetermined concepts. Subsequently, we employ MindSimulator to simulate functional localizer experiments, thereby identifying the concept-selective regions.
>
> We hope that our response addresses your concerns. We are eager to receive your feedback. If you have any further questions or concerns, please do not hesitate to inform us. Thanks for your comments.
>
> Best wishes,
>
> All authors of Submission 180.

---

> ### Comment · Reviewer_KVAs · 2024-11-25
> **I made a mistake. I changed my evaluation**
>
> Based on the discussions presented, along with the points raised by the author of the referenced paper and the explanations provided by the current authors, I acknowledge my initial oversight and have revised the score accordingly. I appreciate the concept explored in this paper and find it both timely and relevant. However, I remain uncertain about the extent of its technical novelty compared to the previous work.

---

> ### Author Response · Authors · 2024-11-29
> **Replying to Reviewer KVAs**
>
> We greatly appreciate your feedback and raising the rating. Regarding your remaining concerns about the novelty of our work, we would like to provide further clarification from three aspects.
>
> `1. Novelty of the encoding model:`
>
> Our proposed fMRI encoding model is an innovative generative model and essentially distinct from previous fMRI encoding models.
>
> First, to the best of our knowledge, previous existing methods are discriminative **regression encoding models**, where the regression model only yields a unique certain fMRI for a given image. However, we argue such an encoding process is different from the nature of human brains. As we mentioned in Section 3.1, observations show that there are noticeable differences in the individual’s brain activity fMRI recordings even when receiving the same visual stimuli. Accordingly, we utilize a generative model to estimate the conditional distribution of p(fMRI|image), guaranteeing the uncertainty of brain activity encoding.
>
> Additionally, we have introduced special designs in our generative encoding model, as outlined below:
>
> - We innovatively trained an fMRI autoencoder to construct the fMRI representation space. Based on this, we were able to learn the conditional distribution in the latent space, which made the training process more stable and resulted in better results.
>
> - We proposed a novel sampling strategy, multi-trials enhancement, which enhances the voxel reproducibility by averaging the fMRI obtained from multiple encoding processes. We further introduced correlated noise, which increases the correlation between multiple generated fMRI and reduces their variability, thereby achieving more stable fMRI synthesis.
>
>
> `2. Novelty of the evaluation techniques:`
>
> We introduce a novel and effective semantic evaluation method leveraging well-trained fMRI-to-image decoding models. This approach assumes that the trained decoding model effectively reconstructs semantic information from (synthetic) fMRI data. Specifically, we utilize the decoding model to reconstruct images from synthetic fMRI and employ visual semantic metrics to measure the semantic quality of the reconstructions. The evaluation technique leverages image semantic assessment to indirectly validate and assess the semantic content embodied within synthetic fMRI data, offering a compelling approach to visualize the semantics embedded in synthetic fMRI. Notable, the new evaluation approach is effective and insightful, contributing
> to the field of brain activity encoding.
>
> `3. Novelty of the concept localization techniques:`
>
> Our proposed encoding-based concept localization technique is powerful and distinct from previous ones. Note that conventional functional localizers (fLoc) that rely on practical experiments of natural stimulus can only localize concepts given by the available stimulus set. In contrast, our approach leveraging the synthetic fMRI capably utilizes arbitrary natural scene images for localization. It has the potential to allow for the localization of any given concept and multi-granular concepts.

---

> > ### Author Response · Authors · 2024-12-01
> > **Kind Reminder for Reviewer KVAs**
> >
> > The author-reviewer discussion phase is approaching its conclusion. We kindly reminder if you have any remaining concerns or questions, please do not hesitate to reach out. We are committed to addressing them to our best. We are looking forward to receiving your feedback. Thank you again for your time and effort.

---

> ### Author Response · Authors · 2024-11-29
> **Replying to Prof. Luo**
>
> Thanks for your constructive response and valuable suggestion. We address each point in detail and make corresponding updates in the revised version.
>
> `Q1`
>
> According to your suggestion, we have changed the colors used in the $R^2$ visualization in Figure 10.
>
> `Q2`
>
> In our experiments, the timestep ratio $\bar{\alpha_T}$ of resting-state fMRI was set to 0.01. We have conducted ablation experiments on the resting-state fMRI, with the results presented below and detailed further in Appendix B.6.
>
> As you expected, the results indeed show that the utilization of resting state initialization does not show a significant improvement. This may be attributed that
>
> 1. the resting state fMRI is not actually the prior state of each sample during the practical image-fMRI data collection;
>
> 2. the generation process during inference does not align exactly with the brain's activity transition from resting states to activated states.
>
> Naturally, it is interesting yet exceedingly difficult to uncover the transition aligning with natural stimulus. Additionally, while our work tries an initial exploration into resting state initialization, it does not constitute the core novelty or contribution of our
> research.
>
> | Methods | Pearson↑ | MSE↓ | PixCorr↑ | SSIM↑ | Alex(2)↑ | Alex(5)↑ | Incep↑ | CLIP↑ | Eff↓ | SwAV↓ |
> | --- | --- | --- | --- | --- | --- | --- | --- | --- | --- | --- |
> | MindSimulator | 0.326 | 0.417 | 0.207 | 0.305 | 90.6% | 97.1% | 92.8% | 89.8% | 0.714 | 0.402 |
> | -w/o resting-state fMRI | 0.322 | 0.418 | 0.204 | 0.302 | 90.9% | 96.5% | 93.0% | 89.8% | 0.712 | 0.407 |
>
> `Q3`
>
> Our objective is to estimate the conditional distribution p(brain∣image), which is, however, highly idealized. In the NSD dataset, each image has only three associated fMRI scans, making accurate estimation challenging due to the limited data. Consequently, the trained diffusion estimator in this setting remains imperfect and requires the addition of correlated noise for further refinement.
>
> We further ablate the correlated noise with 5 independent noises, and the results are shown in the following table. It can be seen that the introduction of correlated noise is beneficial for fMRI encoding. The results confirm our expectation that generating multi-trial synthetic fMRI that are highly correlated with each other can better enhance voxel-wise reproducibility.
>
> We have also added the corresponding results in Appendix B.7.
>
> | Methods | Pearson↑ | MSE↓ | PixCorr↑ | SSIM↑ | Alex(2)↑ | Alex(5)↑ | Incep↑ | CLIP↑ | Eff↓ | SwAV↓ |
> | --- | --- | --- | --- | --- | --- | --- | --- | --- | --- | --- |
> | MindSimulator | 0.326 | 0.417 | 0.207 | 0.305 | 90.6% | 97.1% | 92.8% | 89.8% | 0.714 | 0.402 |
> | -w/o correlated noise | 0.316 | 0.431 | 0.200 | 0.302 | 89.8% | 96.2% | 91.5% | 89.6% | 0.726 | 0.404 |

---

### Official Review · Reviewer_YLHx · 2024-11-04

**Soundness:** 3
**Presentation:** 3
**Contribution:** 3
**Rating:** 6
**Confidence:** 4

**Summary:**

The paper presents a new data-driven approach to localize concept-selective regions in the brain by using synthetic brain recordings generated via a probabilistic model, MindSimulator, conditioned on concept-oriented visual stimuli. This approach enhances coverage and reduces bias, achieving high prediction accuracy in localizing known concept regions.

**Strengths:**

1 - The authors employ a generative fMRI encoding model to synthesize individual fMRI signals corresponding to concept-oriented visual stimuli, addressing the inherent one-to-many correspondence issue between visual stimuli and fMRI recordings.

2 - The paper is well-structured, with a clear formulation of the problem and a thorough description of the proposed model's components and methodology.

3 - The authors provide extensive ablation studies that effectively validate the model architecture's contributions and showcase the performance impact of each component.

**Weaknesses:**

1 - Capturing both spatial and temporal dependencies within the autoencoder is essential for producing meaningful representations of brain activity, which is inherently dynamic. The current model appears to underutilize temporal information based on the description in Supplementary Material Section A. To address this limitation, you might consider adding recurrent layers, such as LSTMs or GRUs, or using 3D convolutions in the autoencoder to enhance temporal processing. Additionally, it would be helpful to clarify in the main text or supplementary materials if and how temporal dependencies are integrated in the current approach. This added information would improve understanding of how well the model aligns with the time-varying nature of fMRI data.

2 - It would be helpful if the authors could clarify their voxel selection and masking process, specifically how spatial relationships between neighboring voxels are preserved when creating the autoencoder input. If there is a risk of losing local spatial context, consider alternative approaches, such as using 3D convolutions or patch-based inputs, which may mitigate this issue and maintain spatial continuity within the masked regions.

3 - To improve the evaluation of your results, please include comparisons with specific, relevant works. For example, you may consider applying a connectivity-based parcellation approach (ref are given below) to both the original and synthetic data to examine whether similar visual networks emerge in each case. Including these comparisons would help readers to contextualize the reported metrics and enable a clearer understanding of your model's relative performance and its contributions to the field.

[1] Du, Y., Fu, Z., Sui, J., Gao, S., Xing, Y., Lin, D., ... & Alzheimer's Disease Neuroimaging Initiative. (2020). NeuroMark: An automated and adaptive ICA based pipeline to identify reproducible fMRI markers of brain disorders. NeuroImage: Clinical, 28, 102375.

[2] Vu, T., Laport, F., Yang, H., Calhoun, V. D., & Adalı, T. (2024). Constrained independent vector analysis with reference for multi-subject fMRI analysis. IEEE Transactions on Biomedical Engineering.

4 - To aid in assessing scalability, please provide details on the computational complexity of the model, including training time, memory usage, and the hardware specifications used in your experiments. These details would offer valuable insight into the practical feasibility of implementing your approach in various research or clinical settings.

**Questions:**

1 - Given the brain’s dynamic complexity and somewhat chaotic behavior, generative models offer both benefits and limitations in modeling brain function. Could the authors evaluate the similarity in temporal and spatial gradients between the original and synthetic data to better assess these dynamics?

2 - Additionally, it would be valuable if the authors could quantify the similarity in functional connectivity maps between the original and synthetic data at each timepoint as well.

---

> ### Author Response · Authors · 2024-11-24
> **Author Rebuttal for Reviewer YLHx (Part-1)**
>
> Dear Reviewer YLHx,
>
> Thanks for your constructive comments and valuable advice. We address your concerns point by point.
>
> `W1:`
>
> First of all, we need to point out that the NSD dataset in the experiments **does NOT contain temporal information**. 1) The image-fMRI pairs in the NSD dataset are collected when participants view a **static image**. 2) The collected fMRI corresponding to each image is **averaged along the temporal dimension** and reduced to 3D, to enhance voxel reproducibility.
>
> Although spatial dependencies in fMRI may provide information to enhance fMRI representation, the spatial structure in fMRI is intricate and brings extreme difficulties to brain encoding/decoding, especially fMRI reconstruction. Accordingly, **the latest related studies [1, 2, 3] generally adopt flattened fMRI as input** and focus on improving the design of generative models.
>
> In addition, **we have made some important initial explorations** to incorporate temporal and spatial dependencies in the experiments.
>
> + **Regarding spatial dependencies**: We have explored extracting **voxel tokens (patches)** with **padding and 3D-convolution** on the original **un-flattened fMRI** and added them with cosine/learnable **position encoding**. However, such a design does **NOT** lead to performance improvement on the fMRI autoencoding task. Instead, it introduces additional computational overhead because the computational complexity of Transformer architecture used for voxel tokens is higher than that of MLP used for 1D voxel sequences.
>
> + **Regarding temporal dependencies**: We have considered the temporal dependence of fMRI in our Diffusion Estimator. Specifically, we have used **previous fMRI representations**, together with image embedding, as **conditions** to guide the synthesis of the current fMRI. We have also tried to utilize noised **previous fMRI representations instead of pure noise** as a starting point for the iterative denoising process. However, these attempts do **NOT** work, introducing additional computational overhead and increasing the instability of the convergence process.
>
> Based on the above attempts, our auto-encoding model still flattens fMRI without incorporating spatial and temporal factors. Certainly, integrating temporal and spatial dependencies is extremely challenging in brain encoding/decoding tasks. Nevertheless, it holds great promise and stands to greatly benefit comprehensive brain analysis and our future research focus.
>
> `W2:`
>
> First, we clarify our voxel selection and masking process as follows:
>
> The **standardized fMRI sample is a 3D matrix**. We used *nsdgeneral*, which is a manually delineated **ROI mask** by NSD, represented as a 0/1 matrix, to **dot-multiply** with the raw fMRI data. After masking, the retained voxels are irregularly shaped but possess connectivity in 3D space. Subsequently, we **flattened the 3D masked fMRI into 1D in the order of length-width-height** and then **removed the zero values** (i.e. masked voxel position).
>
> Indeed, the above fMRI preprocessing considers less-to-none spatial context information. However, **our process is widely adopted by plenty of previous work [1-9] and has been verified effective to yield good results in brain encoding/decoding**. Additionally, we explored the method you mentioned (please refer to W1), but it failed to achieve desirable performance in the fMRI auto-encoding task. It may be attributed that spatial dependencies in fMRI are intricate and bring extreme difficulties to fMRI reconstruction. Accordingly, we adopt the above fMRI preprocessing and pay more attention to the design of the generative encoding model and the semantic evaluation of synthetic fMRI in this work. **In our recent work, we are exploring the contribution of the temporal and spatial dependence in fMRI to brain encoding/decoding and will also consider involving more elaborated fMRI embedding methods such as [10-12] to enhance fMRI auto-encoding performance**.

---

> ### Author Response · Authors · 2024-11-24
> **Author Rebuttal for Reviewer YLHx (Part-2)**
>
> `W3:`
>
> Thanks for your suggestions.
> + **Regarding specific baselines**. The **Transformer encoding model is a specific baseline**. It has the same structure and training process as the method presented in the paper [13]. Notably, the encoding model achieves desirable performance (**2nd** place) in the **Algonauts Challenge [14]**, signifying its superiority to the majority of methods employed in the fMRI encoding challenge. Therefore, **the baselines selected for the experiments are representative** and could yield convincing experimental results.
>
> + **Regarding connectivity-based metrics**. Following your suggestions, **we performed further experiments based on the connectivity-based evaluation metrics**. First, we separately computed the functional connectivity maps for both the synthetic and real fMRI data. Specifically, we adopted the ROIs (words, faces, bodies, places, others) provided by NSD and performed dimensionality reduction on the voxels within each ROI using principal component analysis to guarantee the same length of fMRI. Subsequently, we calculated the Pearson correlation among the different regions to generate the functional connectivity maps. The similarity between the functional connectivity maps of the synthetic and real fMRI data was then evaluated using the mean absolute error. The results reported in the following table show that our MindSimulator achieves the best performance, confirming the superiority of our MindSimulator. We have included the results in **Appendix B.2** of the revised version. We will further complete relevant experiments and incorporate these results and corresponding discussion into the main text.
>
> | Methods                    | connectivity-based metrics$\downarrow $ |
> | -------------------------- | -------------------------- |
> | Linear Encoding Model      | 0.127466                   |
> | Transformer Encoding Model | 0.123541                   |
> | MindSimulator              | 0.122895                   |
>
> `W4:`
>
> All of the training was done on **NVIDIA Tesla V100 32GB GPU**. For a single subject, the training is about **12 GPU hours** for the fMRI autoencoder and **20 GPU hours** for the diffusion estimator. During the inference phase, it takes only an average ~300ms to synthesize an fMRI. We have included the details in **Section 4.2** of the revised version.
>
> `Q1:`
>
> #### 1. Regarding spatial gradients:
>
> We compute the **spatial gradients** of each sample in the test set across all directions and then calculate the absolute difference between the ground-truth fMRI and the synthetic fMRI. The average results of each voxel are reported as follows:
>
> | Methods                    | Average Gradient $\downarrow $|
> | -------------------------- | ---------------- |
> | Linear Encoding Model      | 0.536810         |
> | Transformer Encoding Model | 0.543872         |
> | MindSimulator              | 0.538561         |
>
> It can be observed that in terms of spatial gradients, the results of different methods are similar. The performance of MindSimulator is not optimal since we do not impose a special constraint on the spatial gradient. We have included the results in **Appendix B.2** of the revised version. We will further complete relevant experiments and incorporate these results and corresponding discussion into the main text.
>
>
> #### 2. Regarding the temporal gradients:
>
> In the NSD dataset, **different time points correspond to different image-fMRI pairs**. Therefore, the gradient along the temporal dimension does not make sense in evaluating the fMRI encoding performance.
>
> `Q2:`
>
> Our task focuses on the synthetic quality of fMRI, however **different timepoints correspond to different fMRI samples**. Exploring the similarity of functional connectivity maps across the temporal dimension does not make sense for evaluating the encoding accuracy. Certainly, comparing the similarity of functional connectivity maps between synthetic fMRI and ground-truth fMRI is highly significant, and **we have provided the corresponding evaluation results in the response to `W3`**.

---

> ### Author Response · Authors · 2024-11-24
> **Author Rebuttal for Reviewer YLHx (Part-3)**
>
> **Reference**
>
> [1] Zixuan Gong et al. NeuroClips: Towards High-fidelity and Smooth fMRI-to-Video Reconstruction. NeurIPS 2024.
>
> [2] Andrew F. Luo et al. Brain Mapping with Dense Features: Grounding Cortical Semantic Selectivity in Natural Images With Vision Transformers. arXiv:2410.05266.
>
> [3] Paul S. Scotti et al. MindEye2: Shared-Subject Models Enable fMRI-To-Image With 1 Hour of Data. ICML 2024.
>
> [4] Zixuan Gong et al. Lite-Mind: Towards Efficient and Robust Brain Representation Learning. ACM MM 2024.
>
> [5] Shizun Wang et al. MindBridge: A Cross-Subject Brain Decoding Framework. CVPR 2024.
>
> [6] Ruijie Quan et al. Psychometry: An Omnifit Model for Image Reconstruction from Human Brain Activity. CVPR 2024.
>
> [7] Weihao Xia et al. DREAM: Visual Decoding from Reversing Human Visual System. WACV 2024.
>
> [8] Andrew F. Luo et al. Brain Diffusion for Visual Exploration: Cortical Discovery using Large Scale Generative Models. NeurIPS 2023.
>
> [9] Paul S. Scotti et al. Reconstructing the Mind's Eye: fMRI-to-Image with Contrastive Learning and Diffusion Priors. NeurIPS 2023.
>
> [10] Guobin Shen et al. Neuro-Vision to Language: Enhancing Visual Reconstruction and Language Interaction through Brain Recordings. NeurIPS 2024.
>
> [11] Zijiao Chen et al. CinematicMindscapes: High-quality Video Reconstruction from Brain Activity. NeurIPS 2023.
>
> [12] Huzheng Yang et al. Memory Encoding Model. arXiv:2308.01175v1.
>
> [13] Hossein Adeli et al. Predicting brain activity using transformers. bioRxiv.
>
> [14] Alessandro T. Gifford et al. The algonauts project 2023 challenge: How the human brain makes sense of natural scenes. arXiv:2301.03198.
>
> We sincerely appreciate your insightful feedback and thoughtful comments. We hope that our response effectively addresses your concerns. Feel free to engage in further discussions, and your comments are greatly appreciated.
>
> Best wishes,
>
> All authors of Submission 180.

---

> > ### Comment · Reviewer_YLHx · 2024-11-28
> >
> > Thank you for your detailed responses, which provided valuable insights and helped clarify your work. However, I still have a few concerns:
> >
> > As noted in your response, "Our auto-encoding model still flattens fMRI without incorporating spatial and temporal factors," this limitation may affect the novelty of your approach, both in terms of the model architecture and the problem formulation. Additionally, the performance on computed spatial gradients over synthetic data and ground truth appears lower than other baselines, which raises further questions.
> >
> > While I appreciate your efforts in addressing these points, I will maintain my current score given these remaining concerns.

---

> > > ### Author Response · Authors · 2024-12-01
> > > **Respond to Reviewer YLHx**
> > >
> > > We sincerely thank you for participating in the discussion and providing positive feedback. Your suggestions on integrating spatial and temporal dependencies into the fMRI encoding process have truly inspired us, which, although, remains a challenge that has yet to be adequately addressed by the research community. In the final version of our paper, we will provide a detailed explanation of the attempts we have made. Furthermore, moving forward, we will try our best to explore the significant challenge to build more comprehensive fMRI encoding models and achieve new findings in spatial and temporal metrics. Thank you again for your time and effort.

---

### Meta-Review · Area_Chair_dVjw · 2024-12-19

**Metareview:**

The paper introduces "MindSimulator," a generative framework that synthesizes fMRI responses conditioned on visual stimuli using an fMRI autoencoder and a diffusion model, aiming to localize concept-selective brain regions. While the approach is innovative and leverages stochastic modeling to address the variability in fMRI signals, concerns arise regarding insufficient baselines and confounding factors in the analysis.

Weaknesses include unclear temporal dependency integration, lack of robust voxel selection methods, and limited comparison with state-of-the-art approaches. Additional issues include inadequate handling of confounds in concept localization and questionable justification for specific methodological choices like resting-state initialization. Most were addressed by the authors during rebuttal.

Despite these limitations, the strength of the paper lies in its innovative integration of generative modeling and CLIP-based latent representations to simulate and analyze brain activity, achieving high prediction accuracy. This approach is scalable, provides unique insights into neural encoding of visual concepts, and has the potential to significantly advance neuroscience research methodologies. Thus I recommend for acceptance.

**Additional Comments On Reviewer Discussion:**

During the rebuttal period, the authors addressed concerns from three reviewers. Reviewer Atef suggested comparing their method with Grad-CAM-based approaches for flexible concept selection. The authors explained that Grad-CAM was unsuitable due to differences in concept localization and emphasized their focus on synthetic fMRI data. Atef also recommended testing generalization on other datasets, which the authors clarified through out-of-distribution experiments on CIFAR-10/100 and committed to including in the final paper.

Reviewer SRDC raised concerns about the rigor of claims, the impact of irrelevant factors in natural images, baseline methods, and the lack of multiple comparisons correction. The authors revised their claims, provided additional experiments, and implemented Bonferroni correction. Despite improvements, SRDC maintained their original rating due to unresolved issues with resting-state fMRI data and baseline comparisons.

Reviewer YLHx questioned the lack of spatial and temporal dependencies and the novelty of the approach. The authors clarified that temporal information was not in the NSD dataset and that attempts to incorporate dependencies did not improve performance. Despite addressing concerns, YLHx still questioned the novelty and spatial gradient performance, maintaining critique.

In summary, the authors made revisions to address concerns on method comparisons, generalization, statistical rigor, and baseline choices. I consider all these points to be important ones that have been mostly addressed. However, unresolved issues led reviewers to suggest further revisions in the final version.

---

### Decision · Program_Chairs · 2025-01-22

Accept (Poster)